# A climatology of thermodynamic vs. dynamic Arctic wintertime sea ice thickness effects during the CryoSat-2 era

James Anheuser[1], Yinghui Liu[2], and Jeffrey R. Key[2]

[1]Department of Atmospheric and Oceanic Sciences, University of Wisconsin-Madison, Madison, Wisconsin, USA
[2]Center for Satellite Applications and Research, NOAA/NESDIS, Madison, Wisconsin, USA

**Correspondence:** James Anheuser (anheuser@wisc.edu)

**Abstract.** Thermodynamic and dynamic sea ice thickness processes are affected by differing mechanisms in a changing climate. Independent observational datasets of each are essential for model validation and accurate projections of future sea ice conditions. Here, we present a monthly, Arctic basin-wide and 25 km resolution Eulerian estimation of thermodynamic and dynamic effects on wintertime sea ice thickness from 2010-2021. Estimates of thermodynamic growth rate are determined by coupling passive microwave retrieved snow–ice interface temperatures to a simple sea ice thermodynamic model, total growth is calculated from weekly Alfred Wegener Institute (AWI) European Space Agency (ESA) CryoSat-2 and Soil Moisture and Ocean Salinity (SMOS) combination product (CS2SMOS), and dynamic effects are calculated as their difference. The dynamic effects are further separated into advection and residual effects using a sea ice motion dataset. Our results show new detail in these fields and when summed to a basin-wide or regional scale, are in line with previous studies. Across the Arctic, dynamic effects are negative and about one fourth the magnitude of thermodynamic growth. Thermodynamic growth varies from less than 0.1 m month$^{-1}$ in the central Arctic to greater than 0.3 m month$^{-1}$ in the seasonal ice zones. High positive dynamic effects of greater than 0.1 m month$^{-1}$, twice that of thermodynamic growth or more in some areas, are found north of the Canadian Arctic Archipelago where the Transpolar Drift and Beaufort Gyre deposit ice. Strong negative dynamic effects of less than -0.2 m month$^{-1}$ are found where the Transpolar Drift originates, nearly equal to and opposite the thermodynamic effects in these regions. Monthly results compare well with a recent study of the dynamic and thermodynamic effects on sea ice thickness along the Multidisciplinary drifting Observatory for the Study of Arctic Climate (MOSAiC) drift track during the winter of 2019-2020. Couplets of deformation and advection effects with opposite sign are common across the Arctic, with positive advection effects and negative deformation effects found in the Beaufort Sea and negative advection effects and positive deformation effects found in most other regions. The seasonal cycle shows residual deformation effects and overall dynamic effects increasing as the winter season progresses.

## 1 Introduction

Sea ice thickness is affected by processes that fall into two categories—thermodynamic and dynamic. Thermodynamic processes serve to increase or decrease thickness through phase change; dynamic processes serve to redistribute that thickness both horizontally and vertically. Sea ice models account separately for both thermodynamic and dynamic processes in order to

determine ice thickness and predict how it will respond in a changing climate (Thorndike et al., 1975; Zhang and Rothrock, 2001; Hibler, 1980). These models are evaluated against sea ice thickness observations but current state of the art, basin-wide observations capture only overall ice thickness (Markus et al., 2017; Laxon et al., 2013) and are unable to distinguish between thermodynamic and dynamic processes, which are independently affected through different mechanisms in a changing climate. In order to properly predict how sea ice will respond in a changing climate, these independent processes must be individually evaluated within models, requiring independent observations of each.

A few studies have investigated the effects of thermodynamics and dynamics in sea ice observations on a basin-wide scale (Ricker et al., 2021; Holland and Kimura, 2016). These papers calculate changes to sea ice volume or concentration due to dynamics and define the residual between these dynamic effects and overall changes as the sum of thermodynamic and any other effects. Ricker et al. (2021) completed this analysis from a volume perspective. Volumetric sea ice advection is calculated at regional resolution using sea ice motion and thickness products while thermodynamic effects are taken to be the residual between this advection and overall regional volume change from the sea ice thickness product. The authors go on to compare these results to model output and investigate trends in both the model and observed data. The results of the initial partitioning between dynamic and thermodynamic effects show that on a regional spatial scale and in the mean between 2002-2019, thermodynamic effects are larger than dynamic effects, which vary in sign from region to region. However, results on a sub-regional scale or from the Central Arctic region are lacking. Holland and Kimura (2016) investigated the effects of thermodynamics and dynamics on sea ice concentration over the entire annual cycle and in both polar regions. The authors estimated sea ice concentration advection using sea ice motion vectors determined from observed passive microwave brightness temperatures. The residual difference between this advection term and the overall changes in sea ice concentration is taken to be the sum of thermodynamic effects and ridging. The results show that dynamics play a significant role in maintaining the observed sea ice cover. In particular, they found mechanical redistribution to be an important sink for sea ice concentration in the central Arctic.

In addition to these observation based studies, model studies offer useful context for basin-wide analysis. Keen et al. (2021) examined the sea ice volume budget across the models within the World Climate Research Programme (WCRP) Coupled Model Intercomparison Project Phase 6 (CMIP6; Eyring et al., 2016). In the basin-wide multi-model mean taken from 1960-1989, basal thermodynamic growth is found to be the dominant source term within the volume budget, while dynamics serves as a volume sink at -30% of basal thermodynamic growth. Petty et al. (2018) investigated trends in thermodynamic volume growth as represented by the National Center for Atmospheric Research (NCAR) Community Earth System Model Large Ensemble (CESM-LE; Kay et al., 2015) in order to ascertain the impact of reduced fall ice thickness on thermodynamic growth. Indeed, CESM-LE results suggest that in our current climate, thinner fall ice leads to higher thermodynamic growth rates throughout the winter.

The Multidisciplinary drifting Observatory for the Study of Arctic Climate (MOSAiC; Nicolaus et al., 2022) presented an opportunity for partitioning thermodynamic and dynamic growth from a Lagrangian perspective along the Transpolar Drift over a full year from October 2019 to September 2020. von Albedyll et al. (2022) analyzed data from airborne electromagnetic (AEM) surveys and an ice mass balance buoy network to characterize the annual cycle of both dynamic and thermodynamic

sea ice thickness contributions experienced by the ice pack surrounding the MOSAiC drift station. Thermodynamic growth was modeled using ice mass balance buoy temperature profiles and subtracted from overall ice growth captured by the airborne electromagnetic survey data in order to calculate dynamic sea ice effects as a residual. Overall, the dynamics contribution of 0.1 m out of the 1.1 m growth amounts to 10%. Offering a potential window into basin–wide partitioning of thermodynamics versus dynamics, Koo et al. (2021) compared National Aeronautics and Space Administration (NASA) Ice, Cloud and Land Elevation Satellite (ICESat)-2 data collected over the MOSAiC drift station to ice mass balance buoy thicknesses collected during the field experiment. They found that the mode of ICESat-2 derived sea ice thickness over this region represented level ice under the effects of thermodynamics only while mean and median sea ice thickness included sporadic deformation events which increased sea ice thickness under the effects of sea ice dynamics. Comparing the mean and median observations against the mode observations, the authors conclude that dynamics accounted for 35.6% of the mean sea ice thickness increase and 42.6% of the median sea ice thickness increase over a region enclosed by a 50 km radius around the Polarstern research vessel.

Other studies used estimates of sea ice drift vectors to relate dynamics to sea ice thickness growth, again from a Lagrangian perspective. Kwok and Cunningham (2016) calculated shear and divergence terms averaged over a region north of the Canadian Arctic Archipelago using estimated ice motion vectors during winter from 2011 through 2015. These terms and a constant thermodynamic growth term were linearly regressed to overall sea ice thickness change from European Space Agency (ESA) CryoSat-2. This analysis showed that divergence and shear led to 42% to 56% of overall thickness change averaged across the region in question during those winters, with the remaining change due to thermodynamic effects. The sea ice deformation effects of a 2015 winter storm in the Transpolar Drift north of Svalbard were examined by Itkin et al. (2018), who analyzed AEM measurements of thickness before and after the storm during the Norwegian Young Sea ICE (N-ICE2015) expedition. By tracking individual features in the measured sea ice distribution, they were able to relate divergence and shear to changes in sea ice deformation. In multiplying the effects of this single storm by the climatological average of ten to twenty storms per winter, the authors predict 5% to 10% volume increases due to deformation in the region. von Albedyll et al. (2021) also took advantage of AEM thickness measurements and satellite synthetic aperture radar observations of an unusually large polynya north of Greenland in 2018 to determine a relationship between deformation and thickness changes. Over the 65,000 km$^2$ polynya and over the 1 month of analysis, deformation of ice was found to account for an average of 50% of the thickness increase and in some cases as much as 90% of the thickness increase.

Kwok (2006) analyzed RADARSAT Geophysical Processor System (RGPS) sea ice motion vector derived Eulerian estimates of deformation from 1996 to 2000 over a much larger area, though independent of any sea ice thickness measurements. They report that seasonal ice experiences more deformation than multi-year ice, possibly due to its decreased thickness and strength. They also report a decrease in sea ice divergence as the winter growth season progresses, potentially via the same mechanism of increasing thickness and strength. Even without a link to sea ice thickness, the findings of this study allow for the extrapolation of more localized, short term thickness effect results to a larger spatial scale.

Existing studies have yielded either regional and long term or localized and short term partitioning of thermodynamic and dynamic growth, but a large scale and longer term dataset at sub-regional resolution is lacking—especially from a Eulerian perspective for easier comparison to model outputs. In this study we fill this knowledge gap by presenting a monthly, Arctic

basin-wide and 25 km resolution Eulerian estimation of thermodynamic sea ice thickness growth and dynamic, advection and deformation effects on wintertime sea ice thickness. A difficulty inherent to large scale partitioning of thermodynamic and dynamic effect has been large scale characterization of basal thermodynamic growth. The Stefan's Law Integrated Conducted Energy (SLICE) retrieval methodology allows for daily and basin–wide retrieval of wintertime thermodynamic sea ice growth rate using passive microwave brightness temperatures (Anheuser et al., 2022). Here, the retrieved thermodynamic growth rate is used in conjunction with overall sea ice thickness changes from the radar altimeter aboard the ESA CryoSat-2 satellite (Laxon et al., 2013) to estimate dynamic sea ice effects during the CryoSat-2 era beginning in 2010. With overall sea ice thickness growth provided by the Alfred Wegener Institute (AWI) CryoSat-2 and ESA Soil Moisture and Ocean Salinity (SMOS) combination sea ice thickness product (CS2SMOS; Ricker et al., 2017b) and thermodynamic growth provided by SLICE, Arctic basin-wide sea ice thickness changes due to dynamics are calculated as the residual difference between the two. The effects of advection are also estimated using a sea ice motion vector dataset allowing for the calculation of deformation thickness as a residual of overall dynamic thickness effects and advection thickness effects.

## 2  Data

This analysis requires a sea ice thickness dataset, a sea ice motion estimation dataset and a thermodynamic sea ice growth retrieval.

### 2.1  Satellite Sea Ice Thickness

Launched in 2010, the ESA CryoSat-2 satellite carries the SAR/Interferometric Radar Altimeter-2 (SIRAL-2) instrument (Wingham et al., 2006; Laxon et al., 2013). In order to estimate sea ice thickness, altimetry data from the sensor is first converted to freeboard—the distance between sea level and the top of the ice. The freeboard is then converted to sea ice thickness with an assumed sea ice density and snow loading (Laxon et al., 2013). The footprint of SIRAL-2 radar returns is approximated 300 m by 1500 m (Wingham et al., 2006). Gridded sea ice thickness products with varying averaging periods, grid sizing and radar processing procedures are available from the Centre for Polar Observation and Modelling (CPOM) (Tilling et al., 2018), the NASA Goddard Space Flight Center (GSFC) (Kurtz et al., 2014), the Alfred Wegener Institute (Ricker et al., 2014; Hendricks and Ricker, 2020; Ricker et al., 2017a), the NASA Jet Propulsion Laboratory (Kwok and Cunningham, 2015), the ESA Climate Change Initiative (Hendricks et al., 2018) and the Laboratoire d'Études en Géophysique et Océanographie Spatiales Center for Topographic studies of the Ocean and Hydrosphere (Guerreiro et al., 2017).

The ESA Soil Moisture and Ocean Salinity (SMOS) satellite, initially intended for measuring its namesake soil moisture and ocean salinity, carries the Microwave Imaging Radiometer using Aperture Synthesis (MIRAS) instrument. MIRAS measures 35 to 50+ km resolution passive microwave brightness temperatures at 1.4 GHz (Mecklenburg et al., 2012). At this frequency, the penetration depth into sea ice is high, allowing for retrieval of an ice temperature that can drive a radiative transfer model and yield an estimate of ice thickness (Tian-Kunze et al., 2014).

SIRAL-2 is an active instrument with a relatively small footprint, meaning it takes weeks for CryoSat-2 to cover the entire Arctic Ocean basin. On the other hand, SMOS covers the Arctic basin daily. Furthermore, uncertainties in SMOS sea ice thickness measurements are lower than that of CryoSat-2 when measuring ice less than 0.5 m thick. These complementing characteristics create an opportunity for synergy between the satellites. The AWI CS2SMOS sea ice thickness product takes advantage of this synergy (Ricker et al., 2017b). Through an optimal estimation scheme, the CS2SMOS dataset is available at a weekly time resolution and on a 25 km EASE-Grid 2.0. The dataset is particularly well suited for this study as it provides collocated weekly sea ice thickness observations of the entire basin—necessary for calculating basin–wide differences on a weekly basis. Here we use weekly, wintertime CS2SMOS data from 2010 through 2021. Uncertainties in these data are discussed in Section 5.

## 2.2 Ice motion vectors

In order to estimate the effect of sea ice advection on sea ice thickness, the Polar Pathfinder Daily 25 km EASE-Grid Sea Ice Motion Vectors, Version 4 product from the National Snow and Ice Data Center (NSIDC) (Tschudi et al., 2019; Tschudi et al., 2020) was utilized. The product is available from 1978 to present at daily and weekly temporal resolution. Ice motion vectors are estimated individually from cross correlated satellite brightness temperature data from the Advanced Microwave Scanning Radiometer (AMSR)-Earth Observing System (-E), Advanced Very High Resolution Radiometer (AVHRR), Scanning Multichannel Microwave Radiometer (SMMR), SSM/I and SSMIS, along with International Arctic Buoy Program (IABP) buoy locations, and a National Centers for Environmental Protection (NCEP)/National Center for Atmospheric Research (NCAR) wind reanalysis data derived free drift estimate. These individual ice motion estimates are then merged via an optimal estimation scheme. Each data source is included only when available within the life span of the ice motion product, meaning sources vary throughout the record. Motion vectors are not available amongst the Canadian Arctic Archipelago. When compared against IABP buoy location data from between 1988 and 2011, DeRepentigny et al. (2016) found the weekly sea ice motion vectors to have a 7% median error.

## 2.3 Stefan's Law Integrated Conducted Energy

The SLICE methodology drives Stefan's Law (Stefan, 1891; Lepparanta, 1993) with snow–ice interface temperature retrieved with passive microwave brightness temperatures (Kilic et al., 2019) in order to retrieve instantaneous thermodynamic sea ice thickness growth rate. Stefan's Law of simple sea ice thermodynamics states that:

$$f(t, H, \mathbf{x}) = \frac{\kappa_{eff}}{\rho_i L H} \left( T_f - T_{si} \right) - \frac{F_w}{\rho_i L}, \tag{1}$$

where $f(t, H, \mathbf{x})$ is the thermodynamic growth rate, $\rho_i$ is the density of sea ice, $L$ is the latent heat of fusion, $\kappa_{eff}$ is the effective thermal conductivity of sea ice, $H$ is sea ice thickness, $T_f$ is the freezing point of sea water, $T_{si}$ is the snow–ice interface temperature, and $F_w$ is basal heat flux from the liquid sea water to the solid sea ice. Latent heat of fusion and effective thermal conductivity are calculated using a set of equations that accounts for the multi-phase properties of sea ice (Feltham et al., 2006). In the present analysis, sea ice thickness is taken from the CS2SMOS thickness field, sea ice density is taken to be

917 kg m$^{-3}$, and basal flux is a constant 2 W$^{-2}$. SLICE is available daily and basin-wide but does carry three assumptions—heat conduction in the horizontal direction is assumed to be negligible, it is assumed that there is no thermal inertia present in the ice, and it is assumed that there is no internal heat source, such as the absorption of short wave radiation. These assumptions and the snow–ice interface temperature retrieval are only valid during winter, constraining the analysis to 1 November through 31 March. Additionally, SLICE is only viable in areas with greater than 95% sea ice concentration due to the effects of open water on passive microwave emissivity. SLICE utilizes daily Level 3 passive microwave brightness temperatures and sea ice concentration from JAXA AMSR2 and AMSR-E on a 25 km north polar stereographic grid as provided by the NSIDC (Cavalieri et al., 2014; Markus et al., 2018). We linearly interpolate these data from the north polar stereographic grid to the 25 km EASE-Grid 2.0 for our analysis. More information regarding SLICE can be found in Anheuser et al. (2022).

## 3  Methods

Sea ice thickness is affected by thermodynamic processes and dynamic processes. Thermodynamic processes serve to change sea ice thickness through molecular phase change and dynamic process serve to change local sea ice thickness through the mechanical processes of advection and deformation (ridging or lead formation). An Eulerian governing equation for sea ice thickness sums thermodynamic and dynamic processes:

$$\frac{\partial H}{\partial t} = f(t, H, \mathbf{x}) - \nabla \cdot (\mathbf{u}H), \tag{2}$$

where $H$ is plane slab sea ice thickness; $t$ is time; $f$ is a function of time, thickness and position vector $\mathbf{x}$ describing thermodynamic sea ice thickness increase; and $\mathbf{u}$ is the ice motion vector. The second term on the right hand side represents dynamic sea ice thickness processes.

We aimed to partition weekly basin-wide observations of overall changes to the sea ice thickness field, $\frac{\partial H}{\partial t}$, into its components thermodynamic growth, $f(t, H, \mathbf{x})$, and dynamic effects, $-\nabla \cdot (\mathbf{u}H)$. The result is basin-wide estimates of thermodynamic and dynamic process effects on sea ice thickness. At each weekly time step, $\frac{\partial H}{\partial t}$ was calculated in centered-difference fashion as half the difference between the CS2SMOS sea ice thickness field from the previous time step and that from the following time step. We estimated dynamics, $-\nabla \cdot (\mathbf{u}H)$, as the residual difference between total growth, $\frac{\partial H}{\partial t}$, and expected thermodynamic growth, $f(t, H, \mathbf{x})$, retrieved using the SLICE methodology with the CS2SMOS sea ice thickness from the previous time step as the initial thickness. Due to the 95% or greater sea ice concentration constraint on SLICE, only grid cells that meet this condition at a given time step per an AMSR-E/AMSR2 passive microwave sea ice concentration product (Cavalieri et al., 2014; Markus et al., 2018) are included in this analysis. The results shown are time averages over various time periods. Grid cells that do not contain 95% sea ice concentration for over 40% of the time steps within the averaging time period are discarded and not considered. Figure 1a shows the portion of total time steps that each grid cell shows over 95% sea ice concentration. Also shown in Figure 1 is a map of regions used in Section 4. Regions are defined similar to those shown in Ricker et al. (2021) to aid comparison.

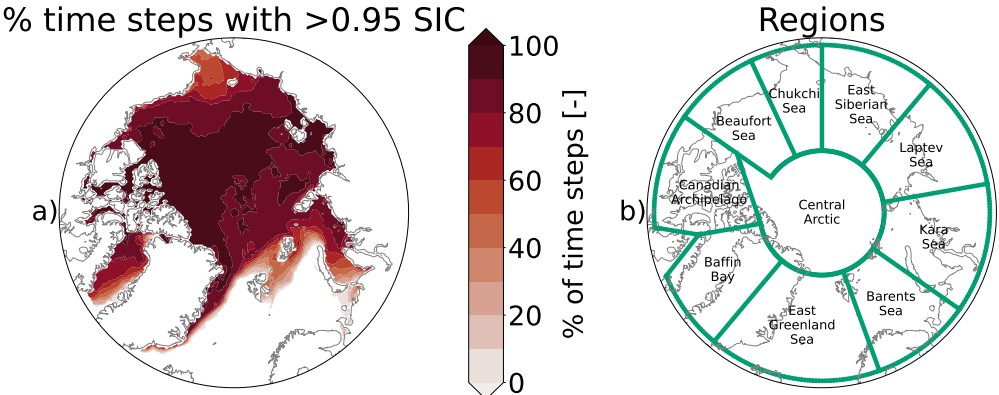

**Figure 1.** Plots showing a) percent of total time steps with 95% or greater sea ice concentration and b) location, extent and corresponding name of regions used in Section 4.

The dynamics term within Eq. 2 can be further decomposed to form:

$$\nabla \cdot (\mathbf{u}H) = (\nabla H) \cdot \mathbf{u} + H(\nabla \cdot \mathbf{u}), \tag{3}$$

where the first term on the right represents changes to local sea ice thickness due to advection, i.e., the movement of ice transporting ice of a new thickness into a grid cell, and the second term on the right represents sea ice thickness changes due to deformation processes caused by divergence of the ice motion vector field. Deformation effect does not include advection and therefore can be considered Lagrangian dynamics—i.e., the dynamic effect as observed by a Lagrangian drifter. Our estimates of dynamic effects were partitioned into advection effects and residual effects using the sea ice motion vector product. At every weekly time step, the first term on the right hand side of Eq. 3 is calculated as the mean of the three time steps centered on the current time step (in order to maintain temporal resolution with the centered difference schemed used to calculated overall thickness change) using the CS2SMOS ice thickness field and sea ice motion vectors and is taken to be advection effect. The residual difference of the overall dynamic effect and this advection effect includes the effects of ice deformation and any other effects that are not accounted for in SLICE or the calculation of advection. These additional effects include lateral growth, snow ice formation and any frazil or new ice growth that occurs above 95% sea ice concentration and is not captured within SLICE. The uncertainty in this residual is a summation of the uncertainties in CS2SMOS, SLICE and the advection calculation. This approach was taken over calculating the deformation effect using the motion vectors and advection effect as a residual because motion vector divergence was found to be significantly more noisy than the motion vector fields themselves. As motion vectors are not available within the Canadian Arctic Archipelago, advection effects and the residual effects are not available in this region.

In summary, the governing equation expressed in Eq. 2 is conceptually reconstructed as follows:

$$\frac{\partial(CS2SMOS)}{\partial t} = SLICE + dynamic\ effect, \tag{4}$$

and the dynamic effect is further decomposed into advection and residual effects with Eq. 3 conceptually reconstructed as follows:

$$dynamic\ effect = (\nabla CS2SMOS) \cdot ice\ motion\ vector + deformation\ effect. \tag{5}$$

The results are presented monthly rather than weekly to reduce noise. Weekly integrations of the terms within Eqs. 4 and 5 were summed to monthly temporal resolution. The dataset covers November through April, due to SLICE assumptions described in Section 2.3 and availability of CS2SMOS data, spanning the ten winters beginning in the years 2010 and 2012-2020. The 2010 data begins on 15 November rather than 1 November along with the availability of CryoSat-2 data and the winter beginning in 2011 is not included due to a gap between availability of passive microwave data from the earlier AMSR-E and latter AMSR2. At each weekly time step, $i$, the following steps were completed at each point on the 25 km EASE-Grid 2.0:

1. $\frac{\partial(CS2SMOS)}{\partial t} = \frac{1}{2} \cdot (CS2SMOS_{i+1} - CS2SMOS_{i-1})$ \hfill (6)

2. $thermodynamic\ growth = SLICE_{i-1}$ \hfill (7)

3. $dynamic\ effect = \frac{\partial(CS2SMOS)}{\partial t} - thermodynamic\ growth$ \hfill (8)

4. $advection\ effect = \sum_{n=i-1}^{i+1} \frac{(\nabla CS2SMOS_n) \cdot ice\ motion\ vector_n}{3}$ \hfill (9)

5. $residual\ effect = dynamic\ effect - advection\ effect$ \hfill (10)

## 3.1 Uncertainty

Uncertainty in the individual weekly observations of thermodynamic, dynamic, advection and residual effect can be calculated using a general formula for uncertainty in a function of several variables (Taylor, 1982):

$$\delta_q = \sqrt{\left(\frac{\partial q}{\partial x}\delta_x\right)^2 + \cdots + \left(\frac{\partial q}{\partial z}\delta_z\right)^2}, \tag{11}$$

where $q$ is the computed value; $x, \cdots, z$ are independent and random inputs to that computed value and $\delta_x, \cdots, \delta_z$ are those inputs associated uncertainties. Applying Eq. 11 to the terms as described in Section 3 and adding covariance terms to the

uncertainty of the space and time derivatives of CS2SMOS, we have:

$$\delta_{thm} = \sqrt{\delta_{SLICE}^2 + \left( \frac{thermodynamic\ growth}{CS2SMOS} \delta_{CS2SMOS} \right)^2} \tag{12}$$

$$\delta_{dyn} = \sqrt{\frac{1}{\Delta t^2} \left( \delta_{CS2SMOS,i-1}^2 + \delta_{CS2SMOS,i+1}^2 - 2 \cdot 0.6 \cdot \delta_{CS2SMOS,i-1} \cdot \delta_{CS2SMOS,i+1} \right) + \delta_{thm}^2} \tag{13}$$

$$\delta_{adv} =$$

$$\sqrt{ \left( \frac{1}{3} \right)^2 \sum_{n=i-1}^{i+1} \left( \frac{u_n}{\Delta x} \sqrt{0.8} \delta_{CS2SMOS,n} \right)^2 + \left( \frac{\partial CS2SMOS_n}{\partial x} \delta_u \right)^2 + \left( \frac{v}{\Delta y} \sqrt{0.8} \delta_{CS2SMOS,n} \right)^2 + \left( \frac{\partial CS2SMOS_n}{\partial y} \delta_v \right)^2 }$$

$$\tag{14}$$

$\delta_{res} = \sqrt{\delta_{dyn}^2 + \delta_{adv}^2}$,          (15)

where $\delta_{thm}$, $\delta_{dyn}$, $\delta_{adv}$, $\delta_{res}$, $\delta_{SLICE}$, $\delta_{CS2SMOS}$, $\delta_u$, and $\delta_v$ are uncertainties in the thermodynamic growth, dynamic effect, advection effect, residual effect, SLICE, CS2SMOS thickness, $x$ direction component of sea ice motion vector, and $y$ direction component of sea ice motion vector, respectively; $u$ is the $x$ direction component of sea ice motion vector; $v$ is the $y$ direction component of sea ice motion vector; $\Delta t$ is time step size; and $\Delta x$ and $\Delta y$ are the grid box size.

The uncertainty in the space and time derivatives of CS2SMOS contain covariance terms. CS2SMOS uncertainty is a significant source of uncertainty within the uncertainty framework above but some portion of this uncertainty would cancel when a difference between time steps or neighboring grid points is taken. Though these covariances have not been explored in literature in relation to CS2SMOS, we look to Fig. 7 within Ricker et al. (2017b) for guidance on correlation between grid cells within a single CS2SMOS field. For the example region depicted in this figure, correlations between thickness observations at
grid points located less than 100 km apart are nearly always greater than 0.6. This 100 km radius includes neighboring grid points which are separated by 25 km and displacement during the two weeks between time steps in Eq. 6, which typically will not exceed 100 km. Based on this figure, we assume a correlation between time steps or neighboring grid points of 0.6 as a conservative estimate. Other than in this instance, our uncertainty formulas do not account for covariances between the input terms. Though covariances may be present across the input data, inclusion of their effects on uncertainty is outside the scope
of this work.

    The uncertainty in SLICE is taken from Anheuser et al. (2022), who report SLICE to have a thermodynamic growth mean bias of $4 \times 10^{-4}$ m d$^{-1}$ and standard deviation bias of $2.2 \times 10^{-3}$ m d$^{-1}$ when compared against ice mass balance buoy data. Here we use this standard deviation as SLICE uncertainty. The analysis presented in Anheuser et al. (2022) does not include the effect of uncertainty in initial sea ice thickness, so we add the second term on the right side of Eq. 12 to account for the
uncertainty in CS2SMOS sea ice thickness. The uncertainty in CS2SMOS is calculated for each week and available in the data product. Tschudi et al. (2020) lists a maximum ice motion vector error of 0.7 cm s$^{-1}$, which we use here for the uncertainty in the ice motion vector components. Lastly, the time step is one week and grid cell size is 25,000 m. Using these inputs, we calculate uncertainty in the thermodynamic growth, dynamic effect, advection effect, residual effect terms at each time step and grid cell location.

Monthly uncertainty is calculated by summation of the weekly uncertainties within each month using Eq. 11. When the terms are averaged across time, the uncertainty of the mean is reduced through the averaging. Applying Eq. 11 to an averaging operation, we have the following:

$$\delta_{mean} = \sqrt{\left(\frac{1}{N}\delta_1\right)^2 + \cdots + \left(\frac{1}{N}\delta_N\right)^2}, \tag{16}$$

where $\delta_{mean}$ is the uncertainty of the mean; N is the number of samples; and $\delta_1, \cdots, \delta_N$ are the individual uncertainties of each sample.

## 4   Results

Figure 2 shows the wintertime mean total sea ice thickness growth, dynamic effects and thermodynamic growth on Arctic sea ice thickness across the entire 10 year analysis period. Total growth and thermodynamic effects are always positive, though with varying magnitudes, while the magnitude and sign of dynamic effect varies across the Northern Hemisphere sea ice. As expected, thermodynamic thickness growth is highest in the seasonal ice zones, in some areas greater than 0.3 m month$^{-1}$, and inversely proportional to the climatological sea ice thickness, leading to less than 0.1 m month$^{-1}$ of thermodynamic growth in much of the Central Arctic. Dynamic effects decrease sea ice thickness over 63% of the area exhibiting ice during the study period and increase sea ice thickness over the remaining area. An increase in sea ice thickness due to dynamics occurs off the Siberian Coast in the Chukchi Sea where the Beaufort Gyre tends to deposit advected ice and similarly north of the Canadian Arctic Archipelago where both the Beaufort Gyre and Transpolar Drift tend to deposit advected ice. The highest positive dynamic effects of greater than 0.1 m month$^{-1}$ occur just north of the central Canadian Arctic Archipelago and in the Chukchi and East Greenland Seas. A decrease in sea ice thickness due to dynamics, in some areas less than -0.2 m month$^{-1}$, occurs in the coastal regions of the Kara and Laptev Seas from where the transpolar drift tends to remove ice, and similarly in the coastal regions of the Beaufort Sea due to a similar effect of the Beaufort Gyre.

Figure 3 shows wintertime mean advection effect and residual effects across the 10 year analysis period. Negative advection effect dominates the Arctic sea ice, covering 70% of the study area. The exception primarily occurs where the Beaufort Gyre advects thick ice from north of the Canadian Arctic Archipelago to the Beaufort Sea. Here, advection effects greater than 0.2 m month$^{-1}$ can be found. The most significant negative advection effects, less than -0.2 m month$^{-1}$, occur in coastal Laptev Sea. Residual effects are negative over 48% of the study area. The largest residual effects of greater than 0.1 m month$^{-1}$ occur where the ice motion tends to deposit and form ice ridges, north of the central Canadian Arctic Archipelago, in the Chukchi Sea and north of Greenland in the Transpolar Drift. The Barents and Kara Seas are dominated by strong negative deformation effects, some areas with less than -0.3 m month$^{-1}$. Coupled with the positive advection effects in the Beaufort Sea are negative deformation effects in this region.

Figure 4 shows the uncertainty in the mean effects shown in Figs. 2 and 3. Per Eq. 16, uncertainties for mean values across time are reduced through the averaging operation. When reduced in this manner, uncertainty in the thermodynamic effect is small. The dynamic effect uncertainty, which includes the effects of uncertainty in both thermodynamic and CS2SMOS overall

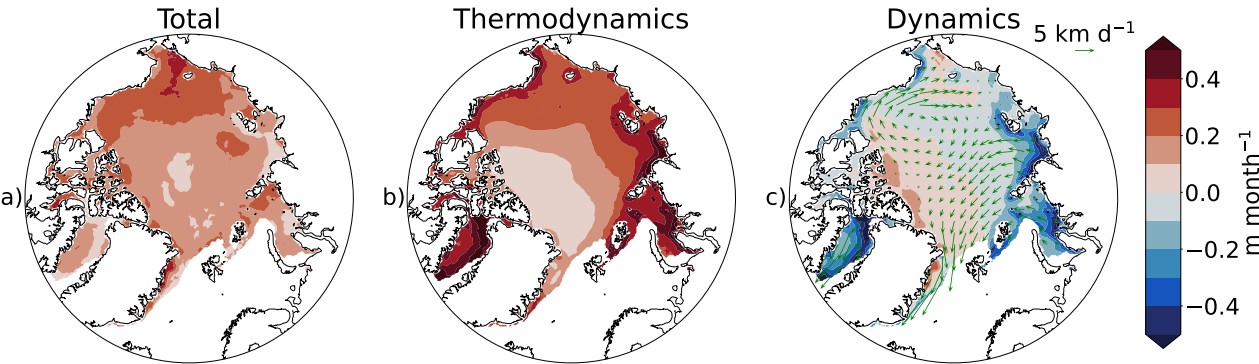

**Figure 2.** Wintertime mean from late 2010 through early 2021 (except the winter of 2011-2012) a) overall sea ice thickness change, b) thermodynamic sea ice thickness effects and c) dynamic sea ice thickness effects. Mean sea ice motion vectors from the same period are also plotted with dynamic effect, which follows spatial patterns suggested by the ice motion vectors.

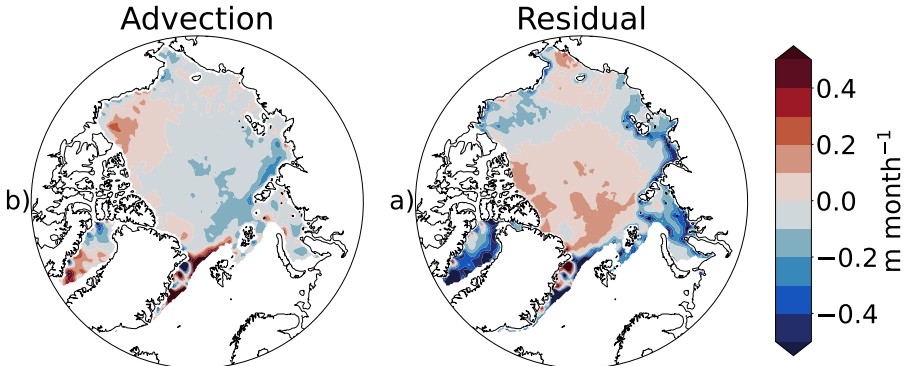

**Figure 3.** Wintertime mean from late 2010 through early 2021 (except the winter of 2011-2012) a) advection sea ice thickness effects and b) residual sea ice thickness effects. Residual effects include those from ice deformation processes.

thickness change is greater than thermodynamic growth uncertainty and highest in the lower latitudes. The uncertainty in CS2SMOS is the larger source of uncertainty in the dynamics term because the differencing of thickness between two separate times steps means overall uncertainty from both time steps are included. This is the case even when the covariance between time steps is included. Advection uncertainty is affected by a similar mechanism as it includes the effects of spatial derivatives in the CS2SMOS field in addition to the motion vector uncertainty. Finally, residual effects have the highest uncertainty as they include the uncertainty of overall change, thermodynamic growth and advection summed in quadrature. In some areas, uncertainty is a similar magnitude to the effects themselves.

To investigate how the budgetary terms interact on a regional scale and facilitate comparison with Ricker et al. (2021) and Keen et al. (2021), Fig. 5 depicts mean total monthly sea ice volume changes due to each effect over the regions shown in Fig. 1 and summed across the entire Arctic. Also included in this figure is mean monthly volume contribution from areas with

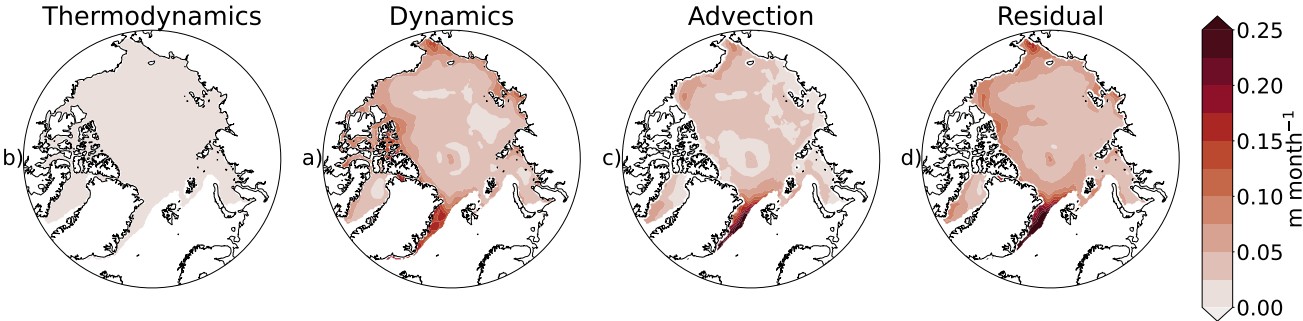

**Figure 4.** Uncertainty calculated per Section for each grid cell during wintertime from late 2010 through early 2021 (except the winter of 2011-2012) sea ice thickness changes due to a) thermodynamic growth, b) dynamic effect, c) advection effects and d) deformation effects. Uncertainty increases with a decrease in latitude as the number of weeks with ice cover and number of satellite overpasses decreases.

less than or equal to the 95% sea ice concentration threshold required for inclusion in the other terms to illustrate the effect of this condition. Each budgetary term was summed across each region and each month before a mean across these months was taken. The $\leq$95% sea ice concentration contribution was calculated by a centered difference scheme similar to Eq. 6 but with the factor of its sea ice concentration included. The uncertainty shown on Fig. 5 is a sum of uncertainties from all grid cells within each region for a given effect which are then binned and summed by month using Eq. 11 and averaged over time using Eq. 16 to yield a monthly uncertainty for each effect and region. Applying Eq. 11 to the uncertainty summation across each region yielded unreasonably low uncertainty, so a basic sum was chosen.

Across the Arctic, thermodynamic growth accounts for $1936 \pm 45$ km$^3$ month$^{-1}$ of volume growth. The dynamic effect across the Arctic is a volume sink of $-583 \pm 359$ km$^3$ month$^{-1}$, which is -30% of thermodynamic growth. Advection accounts for $-126 \pm 229$ km$^3$ month$^{-1}$ of this dynamic effect while residual effects are $-457 \pm 397$ km$^3$ month$^{-1}$. Volume changes in areas that do not have more than 95% ice concentration and therefore are not included in the other terms are $247 \pm 52$ km$^3$ month$^{-1}$ or 13% that of thermodynamic growth. In all regions, thermodynamic growth is the strongest effect. The Central Arctic has the highest thermodynamic volume growth at $320 \pm 12$ km$^3$ month$^{-1}$ and is the only region with a positive dynamic effect at $56 \pm 101$ km$^3$ month$^{-1}$. The regions with the strongest dynamic effects are the Laptev Sea, Kara Sea and Baffin Bay, all regions with significant polynyas, with $-135 \pm 24$ km$^3$ month$^{-1}$, $-142 \pm 22$ km$^3$ month$^{-1}$ and $-162 \pm 19$ km$^3$ month$^{-1}$, respectively. The remaining regions all have negative dynamic effects but they are nearer to zero. The Central Arctic has the strongest advection and residual effects among the regions, with $-106 \pm 77$ km$^3$ month$^{-1}$ advection effect and $163 \pm 127$ km$^3$ month$^{-1}$ residual effect, and is the only region with a positive residual effect. The only regions with a positive advection effect are the Beaufort and East Greenland Seas and Baffin Bay, with $46 \pm 34$ km$^3$ month$^{-1}$, $53 \pm 39$ km$^3$ month$^{-1}$ and $3 \pm 13$ km$^3$ month$^{-1}$, respectively. The remaining regions have negative advection effects that are closer to zero. The strongest negative residual effect is in Baffin Bay, with $-165 \pm 20$ km$^3$ month$^{-1}$. In the Barents and East Greenland Seas, volume contributions in grid cells with less than or equal to 95% sea ice concentration are not insignificant. Other regions have similar contribution from these grid cells below the sea ice concentration threshold but they make up a smaller portion of the total volume changes.

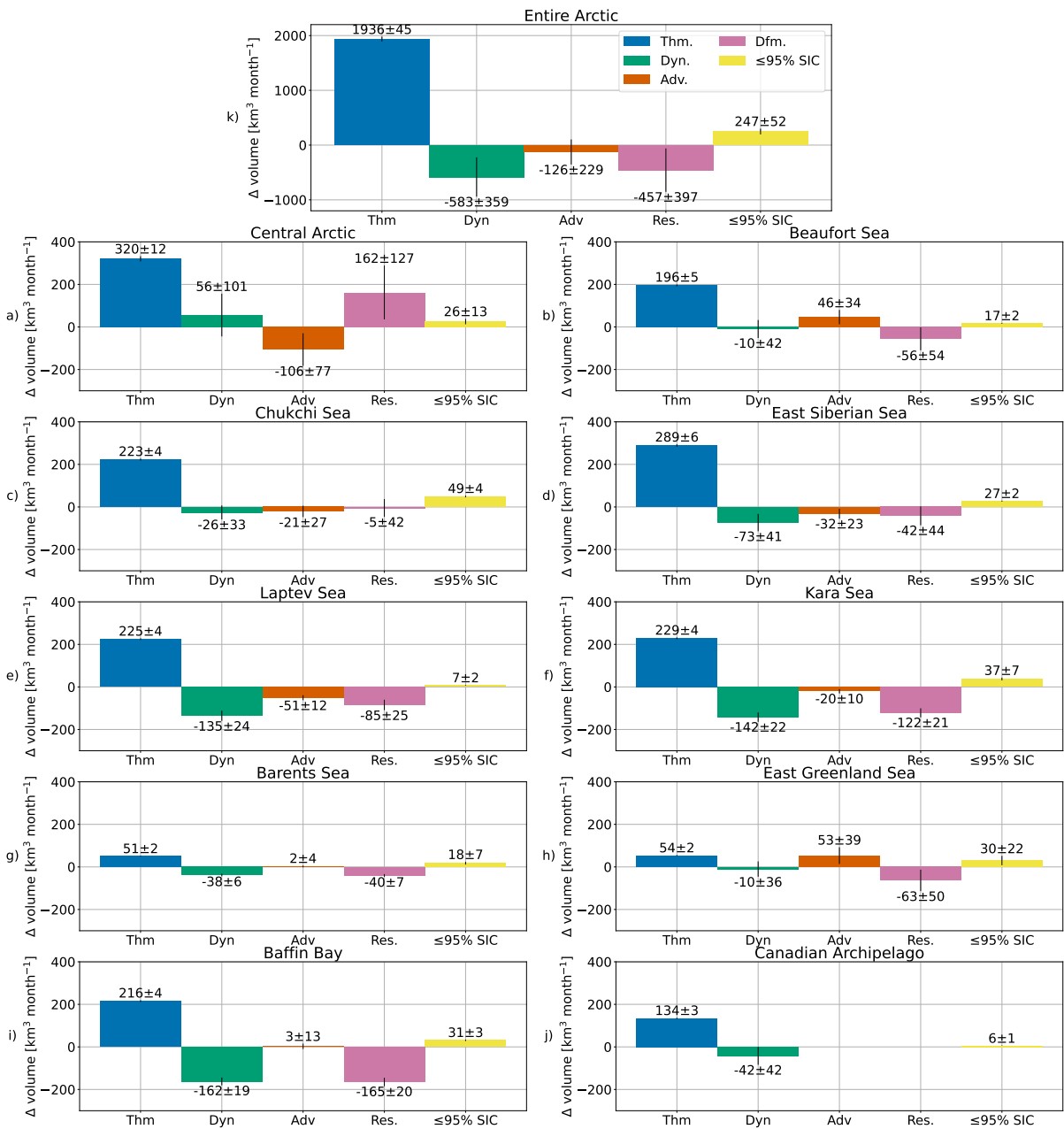

**Figure 5.** Mean monthly volumetric thermodynamic growth (blue), dynamic effect (green), advection effect (orange), deformation effect (pink) and ≤95% sea ice concentration volume changes for a) the entire Arctic, b) Central Arctic, c) Beaufort Sea, d) Chukchi Sea, e) East Siberian Sea, f) Laptev Sea, g) Kara Sea, h) Barents Sea, i) East Greenland Sea, j) Baffin Bay and k) Canadian Archipelago. Over the entire Arctic, dynamic effects are -30% that of thermodynamic growth.

Fig. 6 shows mean dynamics and mean residual effects in relative terms as ratios to thermodynamic growth. To calculate these metrics, the ratio of dynamics and residual to thermodynamics was calculated at each time step and then a time mean of these ratios was taken across all time steps in this study. These plots look similar to those in in Fig. 2 as the scaling quantity, thermodynamic growth, is fairly uniform across most of the Arctic. Nevertheless, the relative importance of dynamics and other residual effects to thermodynamic growth is an important result. Much of the Arctic shows a slightly negative impact

of total dynamics relative to thermodynamic growth. The areas north of the Canadian Arctic Archipelago and Greenland show the highest relative impact of dynamics, with some grid cells showing dynamics with over twice the impact relative to thermodynamics. The coastal regions of the Kara and Laptev Seas show significant negative impacts of dynamics, with magnitudes nearly equal to thermodynamics. The greatest relative importance of residual effects is also found in the areas north of the Canadian Arctic Archipelago and Greenland, though skewing more towards the central Arctic. In these regions,

residual effects can be twice that of thermodynamics. The largest positive relative impact of residual effects is found between Svalbard and Greenland where the Transpolar Drift causes ridging in thick ice that isn't experiencing large thermodynamic growth.

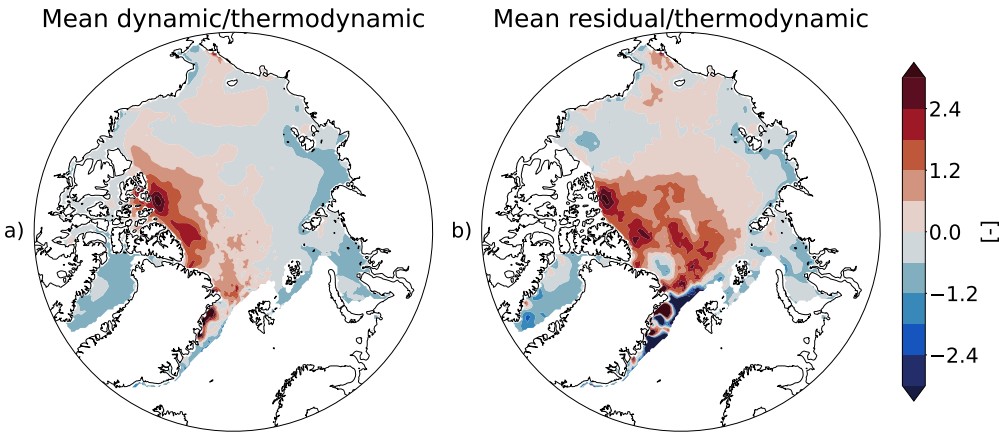

**Figure 6.** Wintertime mean from late 2010 through early 2021 (except the winter of 2011-2012) relative impact of a) dynamic effects over thermodynamic sea ice thickness growth and b) residual effects (excluding advection) over thermodynamic sea ice thickness growth. Alternatively, the figure can be viewed as a) Eulerian dynamics and b) Lagrangian dynamics.

  Fig. 7 shows the monthly mean overall dynamic effect, thermodynamic growth, advection effect and residual effects across the ten winters of data. The thermodynamic growth field, inversely proportional to sea ice thickness, remains consistently low

across the Central Arctic through most of the winter and higher in the perennial ice zones, though growth decreases as thickness increases with time in these areas. The residual and advection fields sum to an overall dynamic effect that increases with time in winter, dominated by the residual effect, potentially as ridging effects increase with overall ice thickness. The monthly residual effects fields depict a negative residual effect in the westward leg of the Beaufort Gyre that peaks in December, decreases in January and is nearly absent in February and March. February and March do, however, depict residual effect maxima north

of Svalbard and Eastern Siberia. Positive advection of ice thickness by the Beaufort Gyre similarly peaks in the early winter, though the broad pattern of the advection field remains consistent throughout the winter.

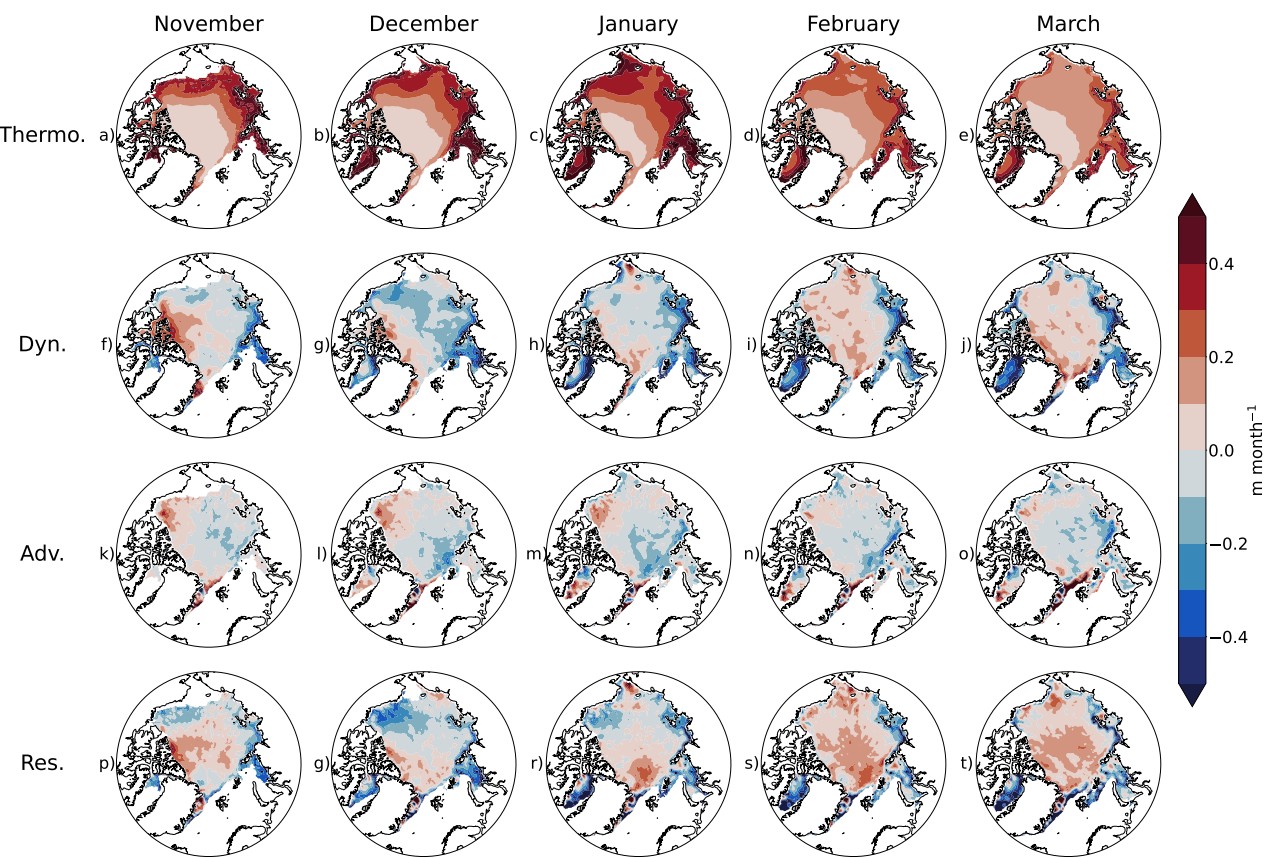

**Figure 7.** Monthly mean a-e) dynamic effect, f-j) thermodynamic effect, k-o) residual effect, and p-t) over the analysis period. Dynamic and residual effects increase through the growth season.

A detailed investigation of interannual variability upon mean thermodynamic, dynamic, advection and deformation effects is outside the scope of this work. However, Fig. 8 shows yearly averages for each of these effects during the study period.

### 4.1  2019-2020 winter and MOSAiC

The MOSAiC field experiment offers an opportunity to drill down further into an individual season's results and compare against similar studies of thermodynamic and dynamic effects along the MOSAiC drift track (Nicolaus et al., 2022). We begin with basin-wide monthly results from the 2019-2020 winter shown in Fig. 9. While the time step of our analysis is one week, the weekly results are noisy and have high uncertainty. Therefore, we present the 2019-2020 results on the monthly time scale. The results show that thermodynamic growth decreases in the peripheral seas throughout the winter while thermodynamic

thickness growth in the central Arctic remains consistently below 0.1 m month$^{-1}$ throughout the year, with both effects due

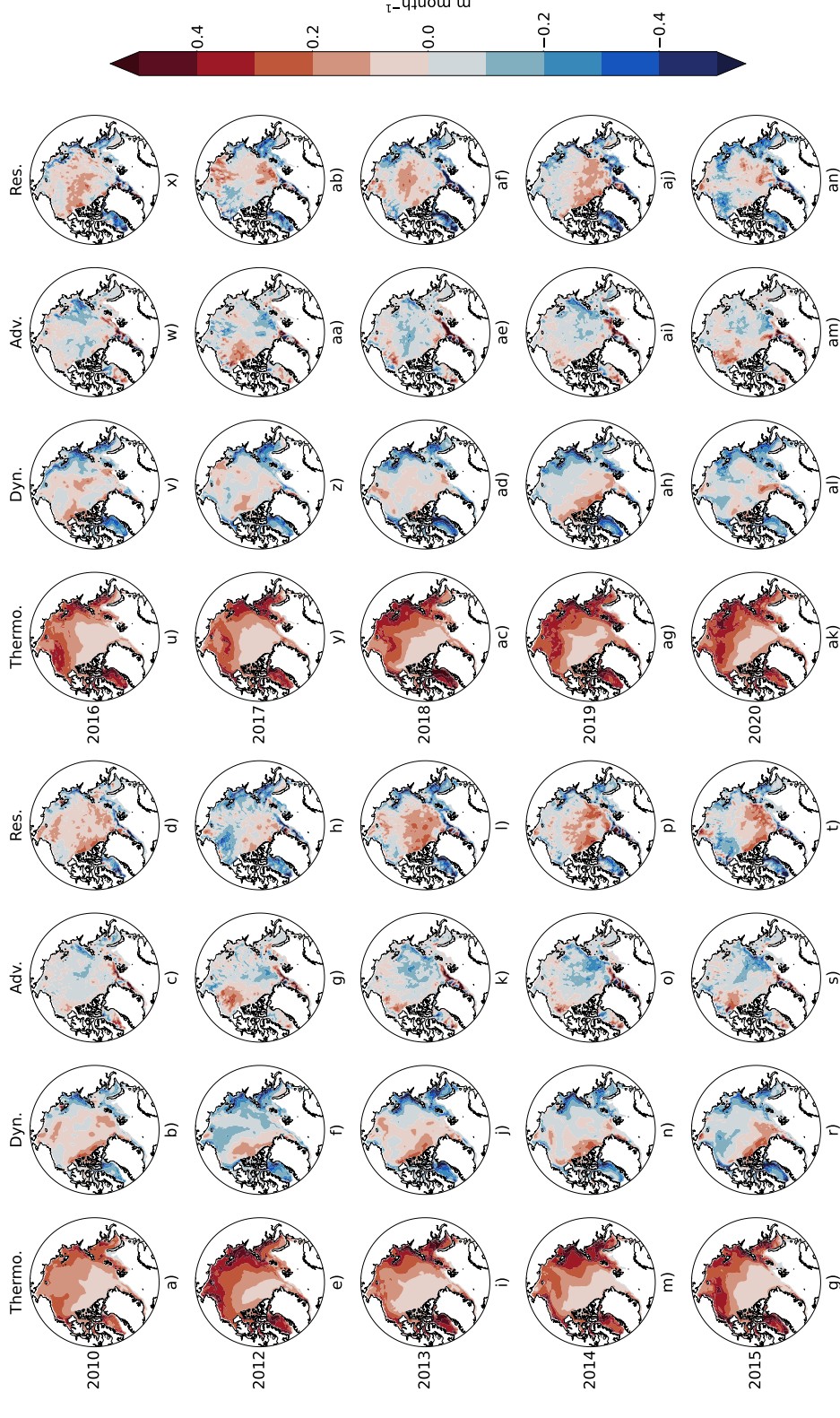

**Figure 8.** Mean thermodynamic, dynamic, advection and residual effects for the winters beginning in 2010-2020 except 2011 with yearly mean ice motion vectors plotted. The patterns of dynamic, advective and residual effects broadly follow the ice motion vectors.

to thickness' inverse relationship to growth rate. The dynamics fields show a shifting area of positive dynamics. In November, positive dynamic effects are climatologically located, north of the Canadian Archipelago. This region of positive dynamics shifts towards Greenland through February, when it increases to over 0.3 m month$^{-1}$ in some areas. In March, a large region of greater than 0.3 m month$^{-1}$ is centered in the Beaufort and Chukchi Seas, while more sparse regions of 0.3 m month$^{-1}$ are still located near the Fram Strait. These patterns are dominated by residual effects during this time. Advection effects appear relatively climatological when compared against Fig. 7, with an area of positive advection effect in the Beaufort Sea beginning in November that decreases in magnitude through the year as negative advection effects grow in the Laptev and Kara Seas, where the Transpolar Drift originates.

Cumulative residual effects (i.e., Lagrangian dynamics), thermodynamics and their sum and relative magnitudes, calculated using the methodology described herein, as experienced by the grid cell nearest to the MOSAiC drift station at each time step, are depicted in Fig. 10 in order to compare these results with those reported by von Albedyll et al. (2022) and Koo et al. (2021). The analysis period is 1 November 2019 through 1 April 2020. Uncertainty in the dynamic and thermodynamic effects are calculated by summing in quadrature the weekly uncertainties within each month using Eq. 11 and then reporting uncertainty by cumulatively summing in quadrature each monthly uncertainty using the same equation. Thermodynamic growth during the period is steady and consistent, ranging between 0.11 m month$^{-1}$ and 0.20 m month$^{-1}$. Cumulative thermodynamic growth at the end of the period is $0.72 \pm 0.03$ m. As expected, Lagrangian dynamic effects were more variable, ranging from -0.06 m month$^{-1}$ to 0.28 m month$^{-1}$. Lagrangian dynamic effects have local maxima in November and February and a cumulative total of $0.67 \pm 0.16$ m. Total growth steadily rises due to the thermodynamic component and follows the shape of the Lagrangian dynamics component. The highest total cumulative growth of 1.39 m is found at the end of the period. As a percentage of total growth, Lagrangian dynamics ends the season at 48% of the total growth. Over a similar study area, Koo et al. (2021) found Lagrangian dynamics to account for 42.6% of mean total growth.

## 5 Discussion

The climatology of ice motion during the CryoSat-2 era as plotted in Fig. 2 suggests the patterns of dynamic effect determined here are sound. The Beaufort Gyre and Transpolar Drift both transport ice towards the Canadian Arctic Archipelago, where high positive dynamic effect are found. The high residual effects in this region are likely explained by ridging. Where the Transpolar Drift originates, in the coastal region of the Laptev Sea, strong negative dynamic effects dominate. In these regions, lead formation likely explains the strongly negative residual effects. Between these regions and the Canadian Arctic Archipelago, a couplet of negative advection effect and positive residual effect characterizes a region where the Transpolar Drift tends to move thinner ice towards thicker ice, all the while experiencing ridging and other effects that increase thickness. Where the motion vectors show the Beaufort Gyre transporting ice westward from north of the Canadian Arctic Archipelago, a similar but reversed couplet of high positive advection effect and strong negative residual effect is found. The ice is transported from a region of climatologically thicker ice north of the Canadian Arctic Archipelago to a region of climatologically thinner ice in the Beaufort Sea, leading to a positive advection effect. In this same region, the Beaufort Gyre flow pattern is diverging

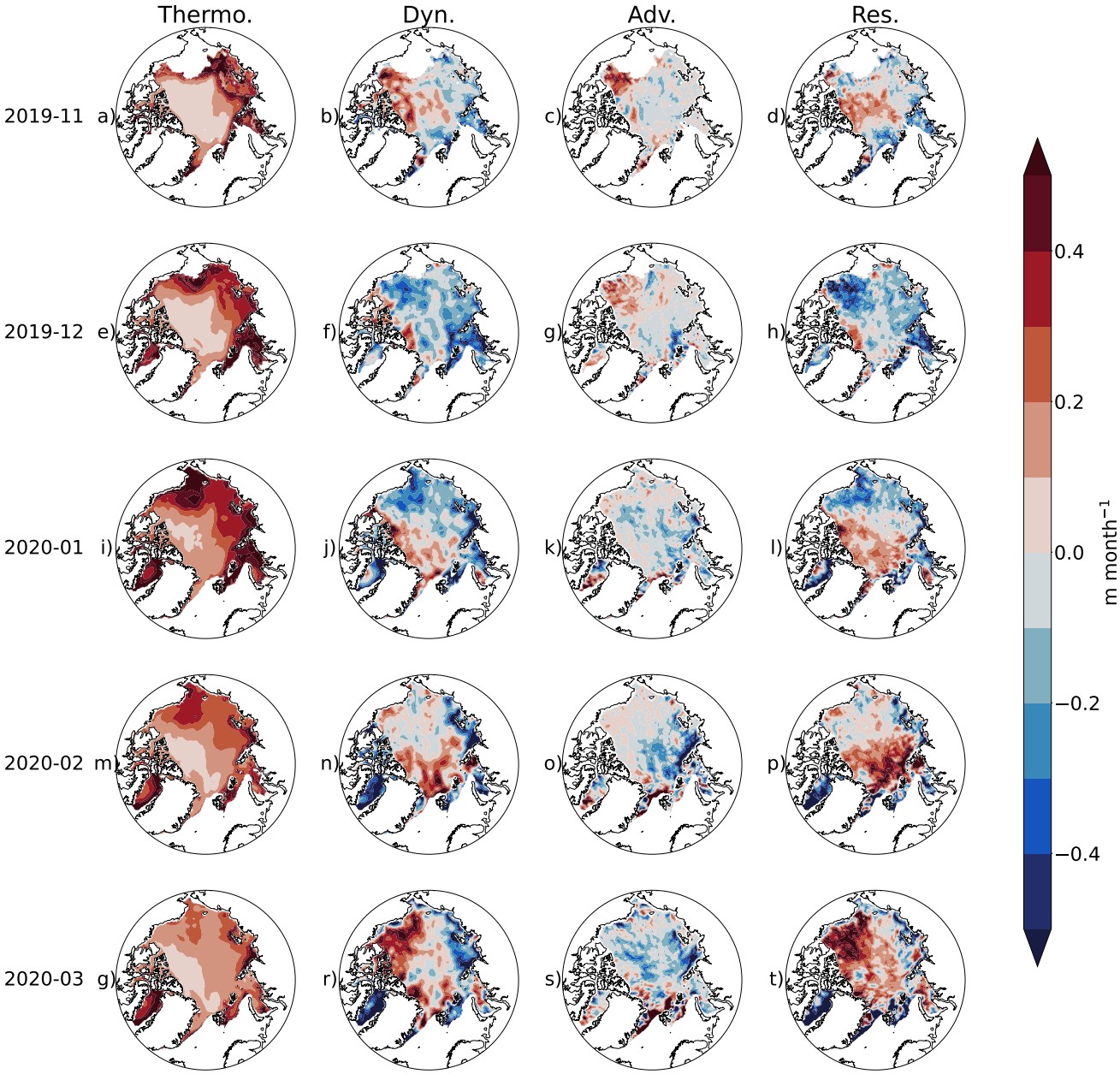

**Figure 9.** Monthly dynamic effect, thermodynamic effect, advection effect and residual effect for a-d) November 2019, e-h) December 2019, i-l) January 2020, m-p) February 2020 and q-t) March 2020. Thermodynamic effect decreases while dynamic and residual effects increase through the growth season.

and accelerating westward. This divergence leads to lead formation and a reduction in mean overall thickness, as reflected by negative residual effects in this region. This matches previous work suggesting high divergence and lead formation in this

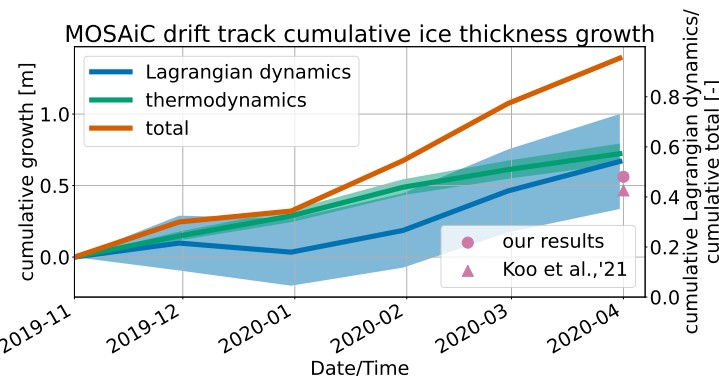

**Figure 10.** Cumulative Lagrangian dynamics (blue line), thermodynamic (green line) and total sea ice thickness growth (orange) from our results on the primary vertical axis and cumulative Lagrangian dynamics over cumulative total sea ice thickness growth during MOSAiC drift track from our results (pink circle) and by Koo et al. (2021) who used ICESat-2 to determine dynamics vs. thermodynamics along the MOSAiC drift track. Lagrangian dynamics accounts for roughly half of all thickness growth by 1 April 2020.

region (Kwok, 2006; Willmes and Heinemann, 2016; Hoffman et al., 2019). As this ice is further advected westward around the Beaufort Gyre, the advection and residual effects return to near zero before reaching areas north of Eastern Siberia where ridging may explain positive dynamic and residual effects. The location and magnitude of these patterns varies year to year with sea ice flow patterns and likely atmospheric conditions as demonstrated in Fig. 8.

Our results add spatial detail to previous studies and agree well with previous studies when summed to similar scales. When summed across the Arctic, our results show dynamic effects are a volume sink with a magnitude that is 30% of thermodynamic growth. This similar to the CMIP6 multi-model mean of 30% as reported by Keen et al. (2021). Volume changes that occurred in grid cells containing sea ice concentration beneath the 95% threshold for inclusion in our results account for 13% of total volume growth, potentially containing much of the frazil growth making up 19% of total growth in the CMIP6 models (Keen

et al., 2021). When our results are summed to the regional scale, they agree well with the previous regional scale results from Ricker et al. (2021), though their results go back to 2002 whereas ours begin in 2018. For each of the six regions included in Ricker et al. (2021), mean thermodynamic growth from our results are within $50\,\mathrm{km}^3\,\mathrm{month}^{-1}$. Of these six regions, our results also agree on which have the strongest dynamic effects (Laptev and Kara Seas) while the rest of the regions show dynamic effects closer to zero–though sometimes with differing signs, perhaps due to the differing time frames.

Kwok (2006) reported on the spatial and seasonal characteristics of Arctic sea ice deformation in the ice motion vector fields using high resolution RGPS data from 1997-2000. Though the years in question do not overlap with the analysis period shown here, their results offer context for understanding dynamic sea ice effects. Their analysis showed divergence in the Beaufort Sea and convergence north of eastern Siberia, a pattern reflected here by the residual effects in these same regions. They show that the fraction of deformed ice in these regions decreases over the course of the growth season, a phenomenon

also shown by a lessening of the negative deformation effects from November through March in Fig. 7. These points support our supposition that residual effects are comprised mostly of the effects of deformed ice on sea ice thickness, whether it be

through ridging (positive residual effects) or lead formation (negative residual effects). The ice motion vectors used here are not suited for vector calculus calculations of deformation terms but a future comparison between concurrent observations of vector deformation fields and dynamics would be fruitful for model improvement. We set a 95% sea ice concentration threshold, at or below which a grid cell is not considered for partitioning of thermodynamic and dynamic changes. In Fig. 5, we report sea ice volume changes that occurred in grid cells with below this threshold to be 266 km$^3$ month$^{-1}$, or 13% that of thermodynamic growth. Volume changes due to changes in sea ice concentration between time steps will be at most 5%, but would be included in the residual effects. It should be noted that new ice formation in leads that have opened up due to divergence in the flow field will not be included in the thermodynamic effect; rather, the balance of the new ice thickness and the thickness of the remaining ice in the grid cell will be quantified as negative residual effect. This type of effect has the potential to be greater than 5%, as the leads may grow to arbitrary size and not be apparent in the passive microwave sea ice concentration product if new ice fills the lead prior to the next passive microwave brightness temperature observation. It is this initial closing of the lead with new ice that will not be captured as thermodynamic growth here.

The mean relative dynamic and residual effects over thermodynamic growth shown in Fig. 6 are useful for understanding the relative importance of these processes and is useful for eventual comparison across time periods. The highest relative impact of dynamics is found north of the Canadian Arctic Archipelago where the Beaufort Gyre and Transpolar Drift both deposit ice and thermodynamic growth is limited due to the high thickness of the ice. The relative residual plot tells a different story. With advection removed, this plot shows that a drifting observer would experience strong negative residual deformation effects relative to thermodynamic growth in the Beaufort Sea and increasingly strong positive residual deformation effects relative to thermodynamics in the Transpolar Drift and eastward leg of the Beaufort Gyre. In the areas north of the Canadian Arctic Archipelago, both dynamics and residual are equal to and greater than thermodynamic effects, consistent with Kwok and Cunningham (2016) who report 42%-56% of mean thickness change is due to deformation. In some areas, the effects of dynamics dominate thermodynamics by a factor of 3. These results confirm the findings of Holland and Kimura (2016) and Ricker et al. (2021) that dynamics play an important role in shaping the climatological sea ice thickness patterns in the Arctic.

Within the methodology used here, the dynamic effects term represents Eulerian dynamics–i.e., a spatially stationary observer of sea ice thickness would observe changes due to thermodynamics and changes captured by the dynamics term, which includes advection. A Lagrangian observer, whom is advecting as described by sea ice motion vector, would not experience changes due to this advection. The Lagrangian observer would only experience changes due to the residual in the framework here. In our interpretation, this residual effect is dominated by deformation effects. In this way, our Eulerian deformation term can be considered Lagrangian dynamics. Two Lagrangian studies of dynamics observed along the MOSAiC drift track offer useful context for validating our Eulerian results (von Albedyll et al., 2022; Koo et al., 2021). The comparison is necessarily between our monthly Eulerian deformation term—i.e., Lagrangian dynamics—from closest in space and time to the MOSAiC drift track and dynamics as described within those studies.

von Albedyll et al. (2022) reported 10% dynamic sea ice thickness growth relative to total growth along the MOSAiC drift, lower than the 48% reported here and the 42.6% reported by Koo et al. (2021). A likely primary cause of this discrepancy is related to temporal resolution. von Albedyll et al. (2022) analyzed AEM sea ice thickness distributions across the 50 km buoy

network at the beginning of the growth season and at the end of the growth season, estimating a cumulative thermodynamic growth during the season using buoy thicknesses. Through a phenomenon acknowledged by the authors, this does not account for dynamics affecting thermodynamic growth throughout the growth season. If dynamics were to increase thickness, as our results and those reported by Koo et al. (2021) show did indeed occur along the MOSAiC drift track, thermodynamic growth rate would decrease as shown in Eq. 1. Without accounting for this effect, thermodynamic growth is overestimated and dynamically driven growth is underestimated, due to the later being calculated as a residual. The higher temporal resolutions in this work and Koo et al. (2021) greatly improve—though do not eliminate—this issue. Indeed, von Albedyll et al. (2022) report a cumulative thermodynamic growth of 1 m, whereas we report total thermodynamic growth of 0.72 m. This is significant, especially given von Albedyll et al. (2022) measured growth between 14 October and 17 April, whereas our analysis period is from 1 November to 1 April (this time discrepancy alone may cause differences as well).

Another potential reason for discrepancy between our results and those of von Albedyll et al. (2022) is the higher overall thickness growth measured by CryoSat-2 relative to the AEM measurements. We report a total mean growth along the MOSAiC drift track of 1.39 m relative to 1.1 m from the AEM survey. Indeed, CS2SMOS shows a mean sea ice thickness of 2.52 m on 1 April 2020 at the MOSAiC location relative to 2.2 m from the AEM surveys. This would manifest as an increase in dynamic effect in our analysis, as dynamics is calculated as a residual when thermodynamics are subtracted from total growth. That there are differences here is not surprising. The satellite measurements are gridded and taken from the nearest grid cell to MOSAiC, while the AEM surveys are centered on the MOSAiC buoy array. On the other hand, though the AEM has higher spatial resolution, the coverage over the 50 km buoy network is not complete. The satellite samples a larger area, although not centered exactly on the MOSAiC buoy array. That our results agree better with Koo et al. (2021) is not surprising, given both studies have used satellite sea ice thickness as the primary dataset. Given that our study aims at a temporally and spatially larger scale while these studies are more focused on singular drift track, we can expect differences in results while using these more localized studies to provide context for our larger scale study.

The calculations and analyses carried out here are all performed using satellite data on a 25 km EASE-Grid 2.0. Whereas sea ice processes can occur on much smaller scales, the results on satellite scale are useful for deciphering patterns and trends on an Arctic basin-wide basis. The CS2SMOS sea ice thickness dataset represents mean sea ice thickness within each grid cell. In actuality, thickness over the grid cell is defined by a distribution rather than a single value. However, without having observed local thickness distributions available at each time step, we have omitted thickness distributions and applied the SLICE retrieval using the mean thickness provided by CS2SMOS. It is likely that implementing thickness distributions would augment our results. Given the non-linear and inverse relationship between thickness and thermodynamic growth rate present in Eq. 1, a distribution favoring ice thinner than the mean thickness over that thicker than the mean within a grid cell would increase thermodynamic growth and decrease dynamic effects (and vice versa). Though snow–ice interface temperature is not expected to vary as greatly as sea ice thickness across a 25 km grid cell, passive microwave snow–ice interface temperature retrieval also represents the mean across each grid cell. The dynamic, thermodynamic, advection and residual effects then necessarily represent mean effects over the grid cell area. While few areas within a grid cell will have experienced exactly the effects described by our results, the cell will have experienced these effects on the mean.

Moving through the methodology described in Section 3 and uncertainty calculations shown in Section 3.1, it is apparent that the uncertainties associated with the various input products stack in such a way that uncertainty increases moving from thermodynamic growth with the lowest uncertainty through dynamic effect, advection effect and residual effect with the highest uncertainty. The uncertainty in thermodynamic growth is a summation in quadrature of SLICE uncertainty and that of the input thickness from CS2SMOS. As dynamic effect is a residual between thermodynamic growth and overall sea ice thickness change from CS2SMOS, uncertainty in this term is a summation in quadrature of uncertainties in these terms. The first term under the radical is uncertainty in overall thickness change rate from CS2SMOS as calculated in Eq. 6. The uncertainty in the advection calculation is an application of Eq. 11 to Eq. 9 with uncertainties in the spatial derivatives of CS2SMOS appearing similar to that of the time derivative in Eq. 6. Residual effects, calculated as the difference between dynamic effect and advection effect, is a summation in quadrature of the uncertainties in these terms, meaning the effects of uncertainty in SLICE, CS2SMOS, CS2SMOS temporal and spatial derivatives and ice motion are all included.

When compared against buoy data and using the buoy thickness as the a priori initial thickness, Anheuser et al. (2022) report SLICE to have a thermodynamic growth mean bias of $4 \times 10^{-4}$ m d$^{-1}$ and standard deviation bias of $2.2 \times 10^{-3}$ m d$^{-1}$. The assumption of 2 Wm$^{-2}$ of basal flux from liquid water to solid sea ice leads to additional uncertainty from SLICE. Assuming a density of 917 kg m$^{-2}$ and a latent heat of fusion of 3.32 x 10$^5$ J kg$^{-1}$, each 1 W m$^{-2}$ of basal sensible heat flux from the liquid sea water to solid sea ice is equivalent to a sea ice thermodynamic growth rate of $2.84 \times 10^{-4}$ m d$^{-1}$. If the assumed 2 W m$^{-2}$ basal sensible heat flux were removed, sea ice growth would increase by $5.67 \times 10^{-4}$ m d$^{-1}$ and an increase from 2 W m$^{-2}$ to 10 W m$^{-2}$ would decrease thermodynamic sea ice thickness growth by $2.27 \times 10^{-3}$ m d$^{-1}$. The SLICE thermodynamic growth retrieval also does not account for lateral melt and freeze processes, or any new or frazil ice growth that occurs above 95% sea ice concentration.

Because Anheuser et al. (2022) used buoy thickness as the initial a priori thickness when comparing against buoys, the uncertainty associated with initial input thickness was not accounted for. To account for this, we have added the second term under the radical in Eq. 12 which accounts for CS2SMOS uncertainty per Eq. 11. The AWI CS2SMOS product merges sea ice thickness retrievals from CryoSat-2 and SMOS into a product that contains reduced uncertainties relative to each instrument's products independently (Ricker et al., 2017b). CryoSat-2 uncertainties are highest over thin ice while SMOS uncertainties are highest over thick ice, creating the opportunity for synergy. CryoSat-2 uncertainties are made up of observational uncertainties or noise and systemic uncertainties or bias (Ricker et al., 2014). Observational uncertainties are reduced through spatial averaging on the grid and optimal estimation methodology used to create the CS2SMOS product. While systemic uncertainties effect estimates of absolute thickness, differencing of thickness between time steps removes them from the estimations of various thickness effects calculated in this work. SMOS uncertainties are caused by uncertainties in the input parameters to the energy budget used to estimate sea ice thickness and are especially high over MYI, results from which are removed from the optimal interpolation. The AWI CS2SMOS product provides an uncertainty value that estimates observational uncertainties for each individual estimate at each time step and grid cell which allows our calculations of thermodynamic growth, dynamic effect, advection effect and deformation effect to also have associated uncertainties at each time step and grid cell.

A potential mechanism for error occurs in the relationship between lead frequency and the snow–ice interface retrieval results. Leads and areas of lower sea ice concentrations contain open sea water exposed at the surface. Sea water has significantly lower emissivity in the microwave band than sea ice, therefore reducing passive microwave brightness temperatures in these regions. To the extent that leads or open water cover a grid cell, these lower brightness temperature would then artificially reduce the retrieved snow–ice interface temperature and cause erroneously large thermodynamic growth rates. Via Eq. 4, erroneously high thermodynamic growth without a change to the CS2SMOS estimates leads to erroneously lower dynamic effects. This phenomenon is difficult to spot because negative dynamic effects are expected in regions with high lead frequency. We restrict our analysis to sea ice concentrations of greater than 95% as retrieved by established passive microwave methods. As such, the highest possible open water fraction within a grid cell is 5%. Assuming emissivity of a satellite field of view is a linear sum of scene type emissivities weighted by area fraction, the effect of 5% open water by area on satellite retrieved snow–ice interface temperature can be approximated. Using an approximate open water emmissivity at 6.9 GHz of 0.56 and sea ice emmissivity of 0.98, the emissivity of a 95% sea ice concentration is 0.959. This reduction in emissivity from 0.98 for a 100% sea ice concentration scene equates to a 5.25 K reduction in brightness temperatures for a 250 K snow–ice interface temperature. Propagating this difference through the retrieval algorithm per Kilic et al. (2019) leads to a reduction of retrieved snow–ice interface temperature of 6 K. In a scenario with thin ice and a small temperature gradient across the ice, this difference could be significant.

## 6   Conclusions

Sea ice models, including those contained within global climate models, account for sea ice thickness and volume changes through separate thermodynamic and dynamic processes. These processes are affected by different mechanisms in a changing climate, meaning independent observations of each are essential for model validation. In this study, we present a monthly, Arctic basin-wide and 25km resolution Eulerian estimation of thermodynamic, dynamic, advection and residual effects on wintertime sea ice thickness from 2010-2021. By retrieving thermodynamic sea ice thickness growth via a simple model driven by passive microwave based snow–ice interface temperature observations (Anheuser et al., 2022) and differencing this growth on a weekly basis from overall sea ice thickness growth from a satellite altimeter/passive microwave combination sea ice thickness product (Ricker et al., 2017b), we show new spatial detail in these effects with a spatial resolution beyond the regional studies available to date. Using a sea ice motion product (Tschudi et al., 2020), we also separated the overall dynamic effect into its Eulerian, independent component effects of advection and residual effects.

When summed to a basin-wide total, our results show dynamic effects are a sea ice volume sink with a magnitude that is 30% of thermodynamic growth, similar to the results of a recent model based study (Keen et al., 2021). Regional totals are also in line with a previous estimates (Ricker et al., 2021). However, our sub-regional results show significant local deviations from these basin-wide and regional results. The highest impact of dynamic effect relative to thermodynamic effect is found north of the Canadian Arctic Archipelago, where dynamic effects account for twice and sometimes three times the thickness growth of thermodynamics. Similarly, residual effects are highest relative to thermodynamics in these regions as well, with residual

effects more than doubling thermodynamics here and slightly farther north, near the North Pole. This is likely due to ridging in these regions.

Thermodynamic growth is lowest in the central Arctic, lower than 0.1 m month$^{-1}$, and highest in the seasonal ice zones, often greater than 0.3 m month$^{-1}$. The highest positive dynamic effects of greater than 0.1 m month$^{-1}$ are found north of the Canadian Arctic Archipelago, where the Transpolar Drift and Beaufort Gyre deposit ice. Strong negative dynamic effects of less than -0.2 m month$^{-1}$ are found where the Transpolar Drift originates. The residual and advection effect fields are dominated by couplets with opposite sign between the two. The Beaufort Sea is characterized by positive advection effects of 0.1 m month$^{-1}$ and negative residual effects of similar magnitude, while most other regions are characterized by negative advection effects, sometimes as low as -0.2 m month$^{-1}$ and positive residual effects, often greater than 0.1 m month$^{-1}$. A seasonal cycle is also shown for all thickness effects effects, the most prominent feature of which is an increasing positive residual thickness effect and overall dynamic thickness effect as the winter season progresses. A potential mechanism for this is increasing ice thickness resisting lead formation and making more ice volume available for ridging.

Monthly results compare well with a recent study of the Lagrangian dynamic and thermodynamic effects on sea ice thickness along the Multidisciplinary drifting Observatory for the Study of Arctic Climate (MOSAiC) drift track during the winter of 2019-2020. Where our data shows Lagrangian dynamics accounting for 48% of growth in the grid cells nearest the drifting study area during this time period, (Koo et al., 2021) found similar results of 42.6% over a similar spatial scale. This lends confidence in our larger spatial and temporal scale results.

Next steps for these data include further interrogation of trends and patterns. There may be a relation to atmospheric conditions or patterns like the Arctic Oscillation or trends related to the changing climate. An additional step will be comparison of these results to those given by sea ice and global climate models.

*Code and data availability.* Data used in creation of all figures is available at https://doi.org/10.5281/zenodo.7987917. Code for creation of data and figures is available at https://doi.org/10.5281/zenodo.7987926 and https://github.com/janheuser/thmdyn. The following auxiliary datasets were used and are available at these locations: AMSR-E and AMSR2 brightness temperatures, https://doi.org/10.5067/AMSR-E/AE_SI25.003 and https://doi.org/10.5067/TRUIAL3WPAUP; AMSR-E and AMSR2 SIC, https://doi.org/10.5067/AMSR-E/AE_SI25.003 and https://doi.org/10.5067/TRUIAL3WPAUP; AWI CS2SMOS v203, https://www.meereisportal.de; sea ice motion vectors, https://doi.org/10.5067/INAWUWO7QH7B; MOSAiC drift track, https://doi.pangaea.de/10.1594/PANGAEA.937193;

*Author contributions.* JA completed all analysis and wrote the first draft under guidance from YL and JK. All authors worked together towards a final draft.

*Competing interests.* The authors declare that they have no conflict of interest.

*Acknowledgements.* This work was funded by the National Oceanic and Atmospheric Administration (NOAA) under grant no. NA20NES4320003. The views, opinions, and findings contained in this report are those of the author(s) and should not be construed as an official National Oceanic

580 and Atmospheric Administration or U.S. Government position, policy, or decision. The merging of CryoSat-2 und SMOS data was funded by the ESA project SMOS & CryoSat-2 Sea Ice Data Product Processing and Dissemination Service and data from November 1st, 2010 to April 1st, 2021 were obtained from https://www.meereisportal.de (grant: REKLIM-2013-04).

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
