# Peer review of "A climatology of thermodynamic vs. dynamic Arctic wintertime sea ice thickness effects during the CryoSat-2 era"

_The Cryosphere, 2022_

## Referee Comment (RC1)

This paper documents the estimation of thermodynamic and dynamical components of sea ice volume changes during the Arctic growth season. The authors have combined 3 data sets: the AWI-SMOS weekly sea ice thickness data, the Polar Pathfinder sea ice drift data and the authors new SLICE, brightness temperature derived thermodynamic ice growth estimate data. The final results presented are quantified and thoroughly compared to ice volume change studies. Such observational derived estimates of ice volume change are challenging to produce, and the authors must be commended at the efforts they have made. However, if this study is to be of use to the wider scientific community, much more information needs to be included. Readers of this paper must then be informed of the conceptual consequences of using the data presented. At the moment very limited information is included about the most appropriate interpretation of the presented results. From this reviewers point of view, this study is as much an accuracy assessment of the SLICE data as an estimate of the components of Arctic sea ice thickness change, though this aspect of the paper is mentioned very little. A more thorough description of the issues are described below.

My main issue with this paper is the incorrect definition of uncertainty used that causes misleading claims within the results and discussion section. The equational form of (6) will give the interannual variability of the budget terms, with no information of the observational uncertainty given. When uncertainty is described for these observational estimates, the reader expects to see information included on how much we can trust these estimates based on how well defined the original measurements are. What role does this uncertainty have in the final budget calculations? The discussion section does attempt to discuss the role of observational uncertainty in the calculations of the paper, though these are compared to scarcely believable claims that the SLICE data has an uncertainty of less than 1mm/ per week. While the uncertainty in AWI-SMOS is plotted, no accurate information on how this plot was calculated is given, making it difficult to interpret. Uncertainty measurements are included in the Pathfinder data set, and I suggest that similar plots need including too. Pathfinder uncertainties can often be as high as 30-50%. As the authors are the creators of the SLICE dataset, then I expect them to also include similar plots showing the uncertainty in this data also. Possible covariances within the data are not mentioned at all. This has the potential to be the highest source of uncertainty in the final values presented. While calculating covariances may be beyond the scope of the study, the possibility of the occurrence needs to be presented.

The values given in this paper with claims such as 'error is highest in the East Greenland, Barents and Kara Seas, where motion vectors are largest and most variable' are telling us only about which regions are most variable from week to week, and from year to year. Indeed it may be possible that such regions have lower observational uncertainty than regions given here that are said to have low uncertainty, when the metric supplied by this study are only indicate that these region have a low interannual variability, or indeed constant values over the growth season.

The first main issue leads into the second limitation of this study: the limited plotted data supplied in the paper. Arctic sea ice is widely described to have large interannual variability, for example in minimum extent and volume.  The results presented and discussed in this paper are mostly from  10 year climatologies, that by definition will remove all such

variability. It is thus then difficult to assess the accuracy and role of the different budget components. Each yearly total is shown in an additional plot in the appendix, and a table is supplied quantifying the key regions. No example weekly calculations are presented at all, which is a crucial omission as this is the time scale of the original calculations. To adequately document the calculations performed for this study, and to be of use to modelling studies, time series of the regions presented can be included. An example figure showing the input and output data from a single week will be a very useful inclusion. Uncertainty data for the shorted timescales needs including too. The final data presented in this paper comes from short time scale calculations. Uncertainty in these calculations comes from these timescales also.

Another concern is due to the definition of the budget equations and the interpretation of this definition. There are three main sources of volume change of sea ice used: rate of change of thickness data, the advection of sea ice thickness, and the new brightness temperature derived thermodynamic ice growth. The final deformation-based estimates of ice volume change is given as the product of the three input data sets. This is conceptually fine, and the results of this method will be of use to the scientific community. However, the authors must tell the reader that it is their own interpretation that gives the difference between the input data as deformation. There are many points within the paper where the deformation values are listed alongside the other components as an additional data value. What is a more accurate definition is thus:

The rate of change in volume and advection estimates are compared to a novel source of data on the thermodynamic change in sea ice thickness. There are relatively large differences between these two data, which are presented here. When considering the source in this difference as from sea ice deformation, these results are the consequence.

This is a more accurate and conceptually more useful description for the wider community. As mentioned above, the final deformation values are thus highly dependent on the uncertainty in the input data. The magnitude of the deformation estimates must be compared to the magnitude of any uncertainty.

The previous concern brings up two notable omissions from the data presented in this paper. First the lack of ice concentration data to convert sea ice thickness to sea ice volume, and then the use of the ice drift data to create an estimation of sea ice divergence. The two data sources can then be used to capture wider scale deformation data and compared to the emergent deformation. Indeed, divergence rates are discussed in this paper, but without presented calculations on the deformation rates then this is purely supposition. The authors comment that the use of Pathfinder for vector calculus applications is not appropriate, if this is so then how can the results of this study be accurate?

Specific points:

L3 and 67 This is not the first such study, see Ricker et al. (2021)

L 80 Previous studies using similar methods to those presented in this paper (ricker et al. 2021, Holland et al. 2016) both also use an ice concentration data set too. This data is crucial in high divergence/advection areas (say the Greenland Sea). Ice thickness products using the

Radar Freeboard method (as presented here) will not capture the loss in ice volume due to ice lead opening.

L 153 What is meant exactly here as 'Deformation effect'? Is it the divergence term that is listed as due to deformation in the previous sentence?

Equations 4 and 5 and the list on lines 174 - 178 Please reformat 'dynamic effect' and others using the normal text rather than equation text format.

L 206 This measure of uncertainty appears to be the interannual variability - the standard deviation of measurements over 10 years of weekly data.

L 211 This is a correct reason for uncertainty, but it is equated to a map of variability.

Figure 1. It will be good to see the mean total change in volume on this plot too as it is the main input data for these calculations.

L 239 It is important to express for this experimental data set, that the results obtained in this paper suggest that 'it experiences a decrease in thickness due to deformation via lead formation'. The data presented here does not observe lead formation, so this must be presented as the authors explanation of the observed results. This is especially true for the interpretation of a 10 year climatology, and even more true for budget calculations that do not incorporate ice concentration data, and thus will not capture the spatial divergence of ice within the time derivative section of equation (2). This rephrasing is crucial to perform throughout the paper.

L 243 'as the flow pattern deposits ice and leads to ridging' again this is not directly observed by the data presented and must be presented as the authors intereration.

L 248 'negative effects from both advection and deformation' again misleading. The advection is an emergent signal present in the data. The deformation is entirely from the resultant difference in the data, and its role as deformation is entirely the authors interpretation. This must be expressed as such.

L 249 'location and strength of the Beaufort Gyre' has this been quantified anywhere? Or is it up to the reader to interpret this entirely from the plotted arrow vectors? If it is from the arrow vectors, please indicate this in the text, as this is a highly interpretive method and not at all accurate. There are other studies that presented data on the Beaufort Gyre and may be a useful inclusion here.

L 251 More information on figure 4 is required. What time scale are these plots calculated? The ration taken at weekly time scales, or over the whole 10 years?

Figure 6: More information is required for this figure caption. Which data come from this study and which are from the MOSAiC campaign? What timescale are the various data on?

L 284 Again, ridging is not observed in this study. What is shown here, is entirely an observed change in sea ice volume that cannot be accounted for using advection or thermodynamics growth estimates. This needs to be written as such before the results can be of use.

L 292, If divergence is discussed, then it really needs to be calculated too.

L 296. It is the other way around. Positive deformation emerges, that can be accounted for through ridging.

L 305, why are they not suitable? Has this been tried? Are the results less certain than the estimation presented in this paper?

L 356 fix equation number.

L 366 Can this claim be backed up? I see no discussion on the uncertainty of the SLICE data. As this is a new data sourced, then it needs to be thoroughly analysed for its accuracy and reliability.

L 384 the observational uncertainty here is compared the data variability. These are not the same thing.

Figure 7. How are these values calculated? Is it the mean uncertainty value taken from the data product?

L 393 I find an uncertainty of 1mm/per week for this data source to be somewhat to good to be true.

L 407. So ice concentration data was used in this study? What data is this?

L 417 A calculation of the resultant uncertainty in advection can thus be calculated. This needs to be quantified.
L 442, Needs to be: where the SLICE thermodynamic growth data can only account for (a third possibly) of, on average, the total observed ice thickness change.

L 447 – this data is not yearly. This is the week to week data surely?

L 453 This claim cannot be made as this is the only uncertainty shown. The uncertainty in Pathfinder and SLICE needs to be shown in similar depth.

L 455 this is a comparison between uncertainty and variability and highly misleading.

Appendix - there is no text in the appendix. Add this figure to the main body of text.

Ricker, Robert, Frank Kauker, Axel Schweiger, Stefan Hendricks, Jinlun Zhang, and Stephan Paul. 'Evidence for an Increasing Role of Ocean Heat in Arctic Winter Sea Ice Growth'.

*Journal of Climate* 34, no. 13 (1 July 2021): 5215–27. https://doi.org/10.1175/JCLI-D-20-0848.1.

Holland, Paul R., and Noriaki Kimura. 'Observed Concentration Budgets of Arctic and Antarctic Sea Ice'. *Journal of Climate* 29, no. 14 (15 July 2016): 5241–49. https://doi.org/10.1175/JCLI-D-16-0121.1.

---

## Author Comment (AC1)

**"Thermodynamic vs. dynamic Arctic wintertime sea ice thickness effects" authors' responses to referee #1**

Thank you for the useful and constructive comments. Authors' responses are in red.

This paper documents the estimation of thermodynamic and dynamical components of sea ice volume changes during the Arctic growth season. The authors have combined 3 data sets: the AWI-SMOS weekly sea ice thickness data, the Polar Pathfinder sea ice drift data and the authors new SLICE, brightness temperature derived thermodynamic ice growth estimate data. The final results presented are quantified and thoroughly compared to ice volume change studies. Such observational derived estimates of ice volume change are challenging to produce, and the authors must be commended at the efforts they have made. However, if this study is to be of use to the wider scientific community, much more information needs to be included. Readers of this paper must then be informed of the conceptual consequences of using the data presented. At the moment very limited information is included about the most appropriate interpretation of the presented results. From this reviewers point of view, this study is as much an accuracy assessment of the SLICE data as an estimate of the components of Arctic sea ice thickness change, though this aspect of the paper is mentioned very little. A more thorough description of the issues are described below.

Thank you for the positive comments and constructive feedback. The SLICE methodology was initially validated in Anheuser et al. (2022) and this paper is an implementation of SLICE towards understanding the relative contributions of thermodynamic and dynamic sea ice effects in the Arctic. Favorable comparison between our results and those found in literature bolster confidence in this work. Additionally, we hope the responses below support this view.

My main issue with this paper is the incorrect definition of uncertainty used that causes misleading claims within the results and discussion section. The equational form of (6) will give the interannual variability of the budget terms, with no information of the observational uncertainty given. When uncertainty is described for these observational estimates, the reader expects to see information included on how much we can trust these estimates based on how well defined the original measurements are. What role does this uncertainty have in the final budget calculations? The discussion section does attempt to discuss the role of observational uncertainty in the calculations of the paper, though these are compared to scarcely believable claims that the SLICE data has an uncertainty of less than 1mm/ per week. While the uncertainty in AWI-SMOS is plotted, no accurate information on how this plot was calculated is given, making it difficult to interpret. Uncertainty measurements are included in the Pathfinder data set, and I suggest that similar plots need including too. Pathfinder uncertainties can often be as high as 30-50%. As the authors are the creators of the SLICE dataset, then I expect them to also include similar plots showing the uncertainty in this data also. Possible covariances within the data are not mentioned at all. This has the potential to be the highest source of uncertainty in the final values presented. While calculating covariances may be beyond the scope of the study, the possibility of the occurrence needs to be presented.

*We have replaced our standard error approach with an uncertainty propagation approach. Please see the updated section below. A key aspect to this calculation is that uncertainty is reduced through temporal averaging in creation of the climatologies, similar to regridding of satellite altimeter data as in AWI CS2SMOS. In regards to Polar Pathfinder uncertainties, DeRepentigny et al. (2016) found the weekly sea ice motion vectors to have a 7% error and the Tschudi et al. (2020) lists a maximum ice motion error of 0.7 cm s-1. Lastly, we have updated the discussion section to reflect the new uncertainty section.*

*Uncertainty in the individual weekly observations of thermodynamic, dynamic, advection and deformation effect can be calculated using a general formula for uncertainty in a function of several variables (Taylor, 1982):*

$$\delta_q = \sqrt{\left(\frac{\partial q}{\partial x}\delta_x\right)^2 + \cdots + \left(\frac{\partial q}{\partial z}\delta_z\right)^2}, \tag{1}$$

*where q is the computed value; $x, \cdots, z$ are independent and random inputs to that computed value and $\delta_x, \cdots, \delta_z$ are those inputs associated uncertainties. Applying Eq. 1 to the terms as described in Section 3, we have:*

$$\delta_{thm} = \sqrt{\delta_{SLICE}^2 + \left(\frac{thermodynamic\ growth}{CS2SMOS}\delta_{CS2SMOS}\right)^2} \tag{2}$$

$$\delta_{dyn} = \sqrt{\left(\frac{1}{\Delta t}\sqrt{2}\delta_{CS2SMOS}\right)^2 + \delta_{thm}^2} \tag{3}$$

$$\delta_{adv} = \sqrt{\left(\frac{u}{\Delta x}\sqrt{2}\delta_{CS2SMOS}\right)^2 + \left(\frac{\partial CS2SMOS}{\partial x}\delta_u\right)^2 + \left(\frac{v}{\Delta y}\sqrt{2}\delta_{CS2SMOS}\right)^2 + \left(\frac{\partial CS2SMOS}{\partial y}\delta_v\right)^2} \tag{4}$$

$$\delta_{def} = \sqrt{\delta_{dyn}^2 + \delta_{adv}^2}, \tag{5}$$

*where $\delta_{thm}$, $\delta_{dyn}$, $\delta_{adv}$, $\delta_{def}$, $\delta_{SLICE}$, $\delta_{CS2SMOS}$, $\delta_u$, and $\delta_v$ are uncertainties in the thermodynamic growth, dynamic effect, advection effect, deformation effect, SLICE, CS2SMOS thickness, x direction component of sea ice motion vector, and y direction component of sea ice motion vector, respectively; u is the x direction component of sea ice motion vector; v is the y direction component of sea ice motion vector; $\Delta t$ is time step size; and $\Delta x$ and $\Delta y$ are the grid box size. These uncertainty formulas do not account for covariances between the input terms. Though covariances may be present across the input data, inclusion of their effects on uncertainty is outside the scope of this work.*

*The uncertainty in SLICE is taken from Anheuser et al. (2022), who report SLICE to have a thermodynamic growth mean bias of $4 \times 10^{-4}$ m d$^{-1}$ and standard deviation bias of $2.2 \times 10^{-3}$ m d$^{-1}$ when compared against ice mass balance buoy data. Here we use this standard deviation as SLICE uncertainty. The analysis presented in Anheuser et al. (2022) does not include*

*the effect of uncertainty in initial sea ice thickness, so we add the second term on the right side of manuscript Eq. 3 to account*

*for the uncertainty in CS2SMOS sea ice thickness. Tschudi et al. (2020) lists a maximum ice motion vector error of 0.7 cm s$^{-1}$, which we use here for the uncertainty in the ice motion vector components. The uncertainty in CS2SMOS is calculated for each week and available in the data product. Lastly, the time step is one week and grid cell size is 25,000 m. Using these inputs, we calculate uncertainty in the thermodynamic growth, dynamic effect, advection effect, deformation effect terms at each time step and grid cell location.*

*When the terms are averaged to form Fig. 1, the uncertainties are reduced through the averaging. Applying 1 to an averaging operation, we have the following:*

$$\delta_{mean} = \sqrt{\left(\frac{1}{N}\delta_1\right)^2 + \cdots + \left(\frac{1}{N}\delta_N\right)^2}, \tag{6}$$

*where $\delta_{mean}$ is the uncertainty of the mean; N is the number of samples; and $\delta_1, \cdots, \delta_N$ are the individual uncertainties of each sample. Figure 5 shows the uncertainty of the mean for each of the processes studied here. The effect with the greatest*

*uncertainty is deformation as it is a summation of uncertainties in the other terms due to it being calculated as a residual from those other terms.*

[Figure]

**Figure 1.** *Uncertainty for each grid cell during wintertime from late 2010 through early 2021 (except the winter of 2011-2012) in sea ice thickness changes due to a) dynamic effects, b) thermodynamic effect, c) advection effect and d) deformation effect. Uncertainty increases with a decrease in latitude as the number of weeks with ice cover decreases and deformation has the highest uncertainty due to being calculated as a residual.*

The values given in this paper with claims such as 'error is highest in the East Greenland, Barents and Kara Seas, where motion vectors are largest and most variable' are telling us only about which regions are most variable from week to week, and from year to year. Indeed it may be possible that such regions have lower observational uncertainty than regions given here that are said to have low uncertainty, when the metric supplied by this study are only indicate that these region have a low interannual variability, or indeed constant values over the growth season.

The updated uncertainty methodology should give more support for this discussion and has been updated per the new
uncertainty estimates.

The first main issue leads into the second limitation of this study: the limited plotted data supplied in the paper. Arctic sea ice is widely described to have large interannual variability, for example in minimum extent and volume. The results presented and discussed in this paper are mostly from 10 year climatologies, that by definition will remove all such variability. It is thus
then difficult to assess the accuracy and role of the different budget components. Each yearly total is shown in an additional plot in the appendix, and a table is supplied quantifying the key regions. No example weekly calculations are presented at all, which is a crucial omission as this is the time scale of the original calculations. To adequately document the calculations performed for this study, and to be of use to modelling studies, time series of the regions presented can be included. An example figure showing the input and output data from a single week will be a very useful inclusion. Uncertainty data for the shorted
timescales needs including too. The final data presented in this paper comes from short time scale calculations. Uncertainty in these calculations comes from these timescales also.

The aim of this work is to present a mean climatology over this time period rather than measurements of individual weeks. As discussed in the uncertainty section, the temporal averaging reduces uncertainties while measurements from individual
weeks are much more uncertain. In our judgement, the smallest time step containing useful data is monthly. We have added monthly regional line plots and monthly basin-wide spatial plots from the 2019-2020 winter to the manuscript to pair with the MOSAiC plot (Figs. 2, 3), which we have also changed to monthly resolution rather than weekly to remain consistent (Fig. 4). The monthly data is created by taking the mean of the weekly data found within each month, sorted by the date on the first day of the week.

[Figure]

**Figure 2.** *Mean monthly time series of dynamic effect (blue), thermodynamic growth (red), advection effect (blue dash dot) and deformation effect (blue dotted) from the 2019-2020 winter for a) the East Greenland Sea, b) the Barents Sea, c) the Kara Sea, d) the Laptev Sea, e) the East Siberian Sea, f) the Chukchi Sea, g) the Beaufort Sea, h) the Canadian Islands, i) the Central Arctic and j) the entire Arctic. Thermodynamic growth typically declines through the season as thickness increases and dynamic effect patterns are variable from region to region.*

[Figure]

**Figure 3.** *Monthly dynamic effect, thermodynamic effect, advection effect and deformation effect for a-d) November 2019, e-h) December 2019, i-l) January 2020, m-p) February 2020 and q-t) March 2020. Thermodynamic effect decreases while dynamic and deformation effects increase through the growth season.*

[Figure]

**Figure 4.** *Cumulative dynamic, thermodynamic and total sea ice thickness growth (primary vertical axis) and cumulative dynamic over cumulative total sea ice thickness growth (secondary vertical axis) along the MOSAiC drift track as determined using the methodology described here. The red triangle represents cumulative dynamics over cumulative total growth over a similar area reported by Koo et al. (2021) who used ICESat-2 to determine dynamics vs. thermodynamics along the MOSAiC drift track. Dynamics accounts for over half of all thickness growth by 1 April 2020.*

Another concern is due to the definition of the budget equations and the interpretation of this definition. There are three main sources of volume change of sea ice used: rate of change of thickness data, the advection of sea ice thickness, and the new brightness temperature derived thermodynamic ice growth. The final deformation-based estimates of ice volume change
is given as the product of the three input data sets. This is conceptually fine, and the results of this method will be of use to the scientific community. However, the authors must tell the reader that it is their own interpretation that gives the difference between the input data as deformation. There are many points within the paper where the deformation values are listed alongside the other components as an additional data value. What is a more accurate definition is thus:

The rate of change in volume and advection estimates are compared to a novel source of data on the thermodynamic change in sea ice thickness. There are relatively large differences between these two data, which are presented here. When considering the source in this difference as from sea ice deformation, these results are the consequence.

This is a more accurate and conceptually more useful description for the wider community. As mentioned above, the final deformation values are thus highly dependent on the uncertainty in the input data. The magnitude of the deformation estimates must be compared to the magnitude of any uncertainty.

The authors agree with this critique on a fundamental level. We have added emphasis to the fact that defining the residual as deformation is our own interpretation rather than a direct measurement of deformation itself. One point of disagreement here is that the differences between the overall volume change, advection, and thermodynamic growth are large. All are of the same order of magnitude.

The previous concern brings up two notable omissions from the data presented in this paper. First the lack of ice concentration data to convert sea ice thickness to sea ice volume, and then the use of the ice drift data to create an estimation of sea ice divergence. The two data sources can then be used to capture wider scale deformation data and compared to the emergent deformation. Indeed, divergence rates are discussed in this paper, but without presented calculations on the deformation rates then this is purely supposition. The authors comment that the use of Pathfinder for vector calculus applications is not appropriate, if this is so then how can the results of this study be accurate?

The SLICE methodology is only viable in areas with 95% or greater sea ice concentration due to open water contamination of the passive microwave brightness temperatures and thus our analysis only applies to times when a given grid cell contains higher than this threshold sea ice concentration. While this was mentioned in the discussion and in Anheuser et al. (2022), it was an unintentional omission on our part to not mention this in the data and methodology sections. We have added the below figure showing the percent of the total study time each grid cell is found to meet this criteria and add a 50% threshold in this metric for each grid cell above which results will not be reported. We certainly do discuss divergence as we suspect that to be the cause of the deformation fields and have added a statement that this is a speculative explanation on our part rather than a measurement of divergence itself in the flow field. Long term divergence in the Polar Pathfinder data is found to be noisy, whereas a long term average of the flow itself (and thus advection) does not exhibit the same noise.

Specific points: L3 and 67 This is not the first such study, see Ricker et al. (2021)

Thank you for this reference. It is highly applicable and we added it to the introduction and discussion.

L 80 Previous studies using similar methods to those presented in this paper (Ricker et al., 2021; Holland and Kimura, 2016)

both also use an ice concentration data set too. This data is crucial in high divergence/advection areas (say the Greenland Sea). Ice thickness products using the Radar Freeboard method (as presented here) will not capture the loss in ice volume due to ice lead opening.

[Figure]

**Figure 5.** *Percentage of time steps with greater than 95% sea ice concentration for each grid cell. Much of the study area spends greater than 90% of the study period with over 95% sea ice concentration.*

We have added these references. The sea ice concentration issue is discussed above and we have added this aspect in the methodology.

L 153 What is meant exactly here as 'Deformation effect'? Is it the divergence term that is listed as due to deformation in the previous sentence?

Correct.

Equations 4 and 5 and the list on lines 174 - 178 Please reformat 'dynamic effect' and others using the normal text rather than equation text format.

Done.

L 206 This measure of uncertainty appears to be the interannual variability - the standard deviation of measurements over 10 years of weekly data. Check this. Are interannual variability and uncertainty in the mean related?

The uncertainty and associated discussion have been updated per above.

L 211 This is a correct reason for uncertainty, but it is equated to a map of variability.

This statement is still true even in the new uncertainty methodology and has been updated to reference the new plots.

Figure 1. It will be good to see the mean total change in volume on this plot too as it is the main input data for these calculations.

We have added a plot of this.

L 239 It is important to express for this experimental data set, that the results obtained in this paper suggest that 'it experiences a decrease in thickness due to deformation via lead formation'. The data presented here does not observe lead formation, so this must be presented as the authors explanation of the observed results. This is especially true for the interpretation of a 10 year climatology, and even more true for budget calculations that do not incorporate ice concentration data, and thus will not capture the spatial divergence of ice within the time derivative section of equation (2). This rephrasing is crucial to perform throughout the paper.

We have added an emphasis to the fact that this explanation and others like it are the interpretation of the authors.

L 243 'as the flow pattern deposits ice and leads to ridging' again this is not directly observed by the data presented and must be presented as the authors intereation.

We have added an emphasis to the fact that this explanation and others like it are the interpretation of the authors.

L 248 'negative effects from both advection and deformation' again misleading. The advection is an emergent signal present in the data. The deformation is entirely from the resultant difference in the data, and its role as deformation is entirely the authors interpretation. This must be expressed as such.

We have added an emphasis to the fact that this explanation and others like it are the interpretation of the authors.

L 249 'location and strength of the Beaufort Gyre' has this been quantified anywhere? Or is it up to the reader to interpret this entirely from the plotted arrow vectors? If it is from the arrow vectors, please indicate this in the text, as this is a highly interpretive method and not at all accurate. There are other studies that presented data on the Beaufort Gyre and may be a useful inclusion here.

While making quantitative connections between the strength or location of the Beaufort Gyre and the effects demonstrated here is out of scope of this study, we thought it reasonable to mention this potential connection. We have replaced "the location and strength of the Beaufort Gyre" with "interannual variability".

L 251 More information on figure 4 is required. What time scale are these plots calculated? The ration taken at weekly time scales, or over the whole 10 years?

The caption states this is a wintertime mean for 2010-2021 but we have added this to the body of the text. Our first submission shows the ratio is the ten year mean of dynamic effect over the ten year mean of thermodynamic growth, though it may be 210 more useful to the take the ratio at weekly time scales and then average.

Figure 6: More information is required for this figure caption. Which data come from this study and which are from the MOSAiC campaign? What timescale are the various data on?

All of the data is from the weekly results demonstrated by this paper except the red triangle with is the Koo et al. (2021) result. We have added "determined using the methodology described here" to the first sentence.

L 284 Again, ridging is not observed in this study. What is shown here, is entirely an observed change in sea ice volume that cannot be accounted for using advection or thermodynamics growth estimates. This needs to be written as such before the 220 results can be of use.

We have added an emphasis to the fact that this explanation and others like it are the interpretation of the authors.

L 292, If divergence is discussed, then it really needs to be calculated too.

The divergence field calculated using the Polar Pathfinder vectors is much more noisy than the vectors themselves.

L 296. It is the other way around. Positive deformation emerges, that can be accounted for through ridging.

We have updated this.

L 305, why are they not suitable? Has this been tried? Are the results less certain than the estimation presented in this paper?

We have tried this approach, but noise in the divergence field makes the results less useful.
L 356 fix equation number.

We have updated this.

L 366 Can this claim be backed up? I see no discussion on the uncertainty of the SLICE data. As this is a new data sourced, then it needs to be thoroughly analysed for its accuracy and reliability.

The uncertainty discussion has been updated per above.

L 384 the observational uncertainty here is compared the data variability. These are not the same thing.

The uncertainty discussion has been updated per above.

Figure 7. How are these values calculated? Is it the mean uncertainty value taken from the data product?

The uncertainty discussion has been updated per above.

L 393 I find an uncertainty of 1mm/per week for this data source to be somewhat to good to be true.

The uncertainty discussion has been updated per above.

L 407. So ice concentration data was used in this study? What data is this?

See our response on the sea ice concentration issue above.

L 417 A calculation of the resultant uncertainty in advection can thus be calculated. This needs to be quantified.

The uncertainty discussion has been updated per above.

L 442, Needs to be: where the SLICE thermodynamic growth data can only account for (a third possibly) of, on average, the total observed ice thickness change.

We have added emphasis to the fact that we use residuals here and our explanations are based on our own interpretation of these residuals.

L 447 – this data is not yearly. This is the week to week data surely?

Yes, we have updated this to "weekly".

L 453 This claim cannot be made as this is the only uncertainty shown. The uncertainty in Pathfinder and SLICE needs to be shown in similar depth.

The uncertainty discussion has been updated per above.

L 455 this is a comparison between uncertainty and variability and highly misleading.

The uncertainty discussion has been updated per above.

Appendix - there is no text in the appendix. Add this figure to the main body of text.

We have moved this figure to the results section where it is briefly introduced.

**References**

Anheuser, J., Liu, Y., and Key, J.: A simple model for daily basin-wide thermodynamic sea ice thickness growth retrieval, The Cryosphere, 16, 4403–4421, https://doi.org/10.5194/tc-16-4403-2022, 2022.

DeRepentigny, P., Tremblay, L. B., Newton, R., and Pfirman, S.: Patterns of Sea Ice Retreat in the Transition to a Seasonally Ice-Free Arctic, J. Climate, 29, 6993 – 7008, https://doi.org/10.1175/JCLI-D-15-0733.1, 2016.

Holland, P. R. and Kimura, N.: Observed Concentration Budgets of Arctic and Antarctic Sea Ice, Journal of Climate, 29, 5241–5249, https://doi.org/10.1175/jcli-d-16-0121.1, 2016.

Koo, Y., Lei, R. B., Cheng, Y. B., Cheng, B., Xie, H. J., Hoppmann, M., Kurtz, N. T., Ackley, S. F., and Mestas-Nunez, A. M.: Estimation of thermodynamic and dynamic contributions to sea ice growth in the Central Arctic using ICESat-2 and MOSAiC SIMBA buoy data, Remote Sensing of Environment, 267, https://doi.org/10.1016/j.rse.2021.112730, 2021.

Ricker, R., Kauker, F., Schweiger, A., Hendricks, S., Zhang, J. L., and Paul, S.: Evidence for an Increasing Role of Ocean Heat in Arctic Winter Sea Ice Growth, Journal of Climate, 34, 5215–5227, https://doi.org/10.1175/jcli-d-20-0848.1, 2021.

Taylor, J. R.: An introduction to error analysis: The study of uncertainties in physical measuremenets, University Science Books, 20 Edgehill Rd., Mill Valley, CA 94941, 1982.

Tschudi, M. A., Meier, W. N., and Stewart, J. S.: An enhancement to sea ice motion and age products at the National Snow and Ice Data Center (NSIDC), Cryosphere, 14, 1519–1536, https://doi.org/10.5194/tc-14-1519-2020, 2020.

---

## Author Comment (AC2)

**"Thermodynamic vs. dynamic Arctic wintertime sea ice thickness effects" authors' responses to referee #2**

Thank you for the useful and constructive comments. Authors' responses are in red.

Review of "A climatology of thermodynamic vs. dynamic Arctic wintertime sea ice thickness effects during the CryoSat-2 era" by Anheuser et al.

This paper aims to quantify the components of dynamic and thermodynamic sea ice growth in the Arctic during the winter season from 2010 to 2021. The authors make use of three products: 1) The SLICE model (Stefan's Law Integrated Conducted Energy) providing thermodynamic ice growth and introduced in an earlier paper ("A climatology of thermodynamic vs. dynamic Arctic wintertime sea ice thickness effects during the CryoSat-2 era") by the authors of this study. 2) The AWI CS2SMOS sea ice thickness data set providing weekly sea ice thickness grids for the Arctic, and here used to derive total sea ice thickness growth. And 3) NSIDC Pathfinder sea ice motion data to derive advection of sea ice.

The topic is relevant and the approach using the SLICE model is interesting. In general, I think this paper could be interesting and a benefit for the sea ice and climate community. But from my point of view the study is lacking further information on methods (and may be required corrections), and a sound uncertainty estimation. I have a few major concerns:

Thank you for the positive comments.

The SLICE thermodynamic ice growth is subtracted from the total growth in sea ice thickness using the CS2SMOS product. Are the authors aware that the CS2SMOS sea ice thickness for each grid cell does not include open water? So hypothetically assuming that within a grid cell (with pure level ice) sea ice diverges, forming leads, the averaged ice thickness will be the same in CS2SMOS (especially for the CryoSat-2 domain). In other words, thickness=0, is not used for averaging. But I cannot see that this is considered in the current approach.

The SLICE methodology is only viable over 95% or greater sea ice concentration due to open water contamination of the passive microwave brightness temperatures and thus our analysis only applies to times when a given grid cell contains higher than this threshold sea ice concentration. While this was mentioned in the discussion and in the SLICE paper, it was an unintentional omission on our part to not mention this in the data and methodology sections. This condition means that even without considering the effects of changing sea ice concentration, the maximum error in a given term will be 5%. We have also added the below figure showing the percent of the total study time each grid cell is found to meet this criteria and add a 50% threshold in this metric for each grid cell above which results will not be reported.

[Figure]

% time steps with >0.95 SIC

**Figure 1.** *Percentage of time steps with greater than 95% sea ice concentration for each grid cell. Much of the study area spends greater than 90% of the study period with over 95% sea ice concentration.*

Partly related to 1): I wonder how new ice formation in leads is affecting the overall findings in this study. It is not clear to me how this is handled in this study. SLICE does not seem to consider new ice formation in leads or am I wrong?

You are correct, we have no way of capturing new ice within a gridcell using the methodology of this paper. With a 95% sea ice concentration threshold, this will likely not be a significant source of volume generation. Nonetheless, have added a description of this issue to the discussion.

The way uncertainties are considered in this study does not seem sound, or at least needs further explanation. I would assume that the uncertainty of the climatological mean as calculated here (Eq. 6) is mostly affected by temporal (interannual and seasonal) variability. This needs some improvement from my point of view. Moreover, the SLICE uncertainty for the weekly thermodynamic ice growth in most of the Central Arctic is close to 0, which does not seem realistic, also considering the comparison with independent data sets in Anheuser et al. (2022).

A comprehensive uncertainty analysis is important here, since different input products are used, where either of them adds to the uncertainty budget. Uncertainty of the sea ice motion product is barely mentioned, but especially in the Fram Strait I believe this can lead to significant errors in the final retrievals, also since ice thickness is very heterogenous there.

These are good constructive critiques. We have replaced our standard error approach with an uncertainty propagation ap-proach. Please see the updated section below. A key aspect to this calculation is that uncertainty is reduced through temporal averaging in creation of the climatologies, similar to regridding of satellite altimeter data as in AWI CS2SMOS. In regards to Polar Pathfinder uncertainties, DeRepentigny et al. (2016) found the weekly sea ice motion vectors to have a 7% error and the Tschudi et al. (2020) lists a maximum ice motion error of 0.7 cm s-1. Lastly, we will update the discussion section to reflect the new uncertainty section.

*Uncertainty in the individual weekly observations of thermodynamic, dynamic, advection and deformation effect can be calculated using a general formula for uncertainty in a function of several variables (Taylor, 1982):*

$$\delta_q = \sqrt{\left(\frac{\partial q}{\partial x}\delta_x\right)^2 + \cdots + \left(\frac{\partial q}{\partial z}\delta_z\right)^2}, \tag{1}$$

*where $q$ is the computed value; $x, \cdots, z$ are independent and random inputs to that computed value and $\delta_x, \cdots, \delta_z$ are those inputs associated uncertainties. Applying Eq. 1 to the terms as described in Section 3, we have:*

$$\delta_{thm} = \sqrt{\delta_{SLICE}^2 + \left(\frac{thermodynamic\ growth}{CS2SMOS}\delta_{CS2SMOS}\right)^2} \tag{2}$$

$$\delta_{dyn} = \sqrt{\left(\frac{1}{\Delta t}\sqrt{2}\delta_{CS2SMOS}\right)^2 + \delta_{thm}^2} \tag{3}$$

$$\delta_{adv} = \sqrt{\left(\frac{u}{\Delta x}\sqrt{2}\delta_{CS2SMOS}\right)^2 + \left(\frac{\partial CS2SMOS}{\partial x}\delta_u\right)^2 + \left(\frac{v}{\Delta y}\sqrt{2}\delta_{CS2SMOS}\right)^2 + \left(\frac{\partial CS2SMOS}{\partial y}\delta_v\right)^2} \tag{4}$$

$$\delta_{def} = \sqrt{\delta_{dyn}^2 + \delta_{adv}^2}, \tag{5}$$

*where $\delta_{thm}$, $\delta_{dyn}$, $\delta_{adv}$, $\delta_{def}$, $\delta_{SLICE}$, $\delta_{CS2SMOS}$, $\delta_u$, and $\delta_v$ are uncertainties in the thermodynamic growth, dynamic ef-*
*fect, advection effect, deformation effect, SLICE, CS2SMOS thickness, $x$ direction component of sea ice motion vector, and $y$ direction component of sea ice motion vector, respectively; $u$ is the $x$ direction component of sea ice motion vector; $v$ is the $y$ direction component of sea ice motion vector; $\Delta t$ is time step size; and $\Delta x$ and $\Delta y$ are the grid box size. These uncertainty formulas do not account for covariances between the input terms. Though covariances may be present across the input data, inclusion of their effects on uncertainty is outside the scope of this work.*

*The uncertainty in SLICE is taken from Anheuser et al. (2022), who report SLICE to have a thermodynamic growth mean bias of $4\times10^{-4}$ m d$^{-1}$ and standard deviation bias of $2.2\times10^{-3}$ m d$^{-1}$ when compared against ice mass balance buoy data. Here we use this standard deviation as SLICE uncertainty. The analysis presented in Anheuser et al. (2022) does not include the effect of uncertainty in initial sea ice thickness, so we add the second term on the right side of manuscript Eq. 3 to account for the uncertainty in CS2SMOS sea ice thickness. Tschudi et al. (2020) lists a maximum ice motion vector error of 0.7 cm s$^{-1}$,*
*which we use here for the uncertainty in the ice motion vector components. The uncertainty in CS2SMOS is calculated for each week and available in the data product. Lastly, the time step is one week and grid cell size is 25,000 m. Using these inputs, we calculate uncertainty in the thermodynamic growth, dynamic effect, advection effect, deformation effect terms at each time step and grid cell location.*

     *When the terms are averaged to form Fig. 1, the uncertainties are reduced through the averaging. Applying 1 to an averaging*
*operation, we have the following:*

$$\delta_{mean} = \sqrt{\left(\frac{1}{N}\delta_1\right)^2 + \cdots + \left(\frac{1}{N}\delta_N\right)^2}, \tag{6}$$

*where $\delta_{mean}$ is the uncertainty of the mean; N is the number of samples; and $\delta_1, \cdots, \delta_N$ are the individual uncertainties of each sample. Figure 2 shows the uncertainty of the mean for each of the processes studied here. The effect with the greatest uncertainty is deformation as it is a summation of uncertainties in the other terms due to it being calculated as a residual from those other terms.*

[Figure]

**Figure 2.** *Uncertainty for each grid cell during wintertime from late 2010 through early 2021 (except the winter of 2011-2012) in sea ice thickness changes due to a) dynamic effects, b) thermodynamic effect, c) advection effect and d) deformation effect. Uncertainty increases with a decrease in latitude as the number of weeks with ice cover decreases and deformation has the highest uncertainty due to being calculated as a residual.*

Given these points, I suggest major revisions are needed.

Specific comments: L51 & 55: AEM (airborne electromagnetic) sounding measures the sea ice thickness, not freeboard (can be retrieved only indirectly).

Thank you for pointing this out, we have made this change.

L66: There are some recent studies that already investigated the ice growth components and should be mentioned and cited here: e.g.: Petty et al. (2018), Ricker et al. (2021). The latter also compared observational ice growth retrievals with model outputs.

We have added these references and included a comparison with our results where appropriate.

L73-75: Why is CS2SMOS not mentioned here (and in the entire introduction) as it is used in thus study?

This is a good point. We have updated the section to reference the CS2SMOS product rather than only CryoSat-2.

L98: I suggest providing numbers here for the footprint (e.g. 300 m (along track) x 1600 m (across track)).

We have added this information.

L155: "so to can" ... rewording

We have updated this.

L 174: Would it not be correct to use a three-point linear regression over [i-1,i+1], centered at "i"? Otherwise, the gradient is not centered on the target week "i", but in between "i" and "i+1".

We have tried this methodology and though it doesn't appear to significantly change the results, we have implemented this
in the next submission.

L193-194: "The most significant negative advection effects, less than 0.04 m wk-1, ...". It is misleading when speaking of negative effects but then stating a "less than" a positive number, which could be again a postitive value.

We have updated this wording.

L197: Is negative deformation = lead formation?

Yes, in our interpretation a negative deformation effect is explained by lead formation and new ice filling those leads, thereby
reducing the thickness of the ice within a grid box.

L465: Please state the version number of CS2SMOS. For most recent data and version history: https://spaces.awi.de/display/CS2SMOS/CryoSat-SMOS+Merged+Sea+Ice+Thickness

We have added this information.

Figures 1&2: The upper and lower limits of the color map seem to be saturated in some areas. I suggest to adjust the limits.

We have changed the limits on these plots.

Figure 6: Increase the resolution of the figure.
We have updated this.

**References**

Anheuser, J., Liu, Y., and Key, J.: A simple model for daily basin-wide thermodynamic sea ice thickness growth retrieval, The Cryosphere, 16, 4403–4421, https://doi.org/10.5194/tc-16-4403-2022, 2022.

DeRepentigny, P., Tremblay, L. B., Newton, R., and Pfirman, S.: Patterns of Sea Ice Retreat in the Transition to a Seasonally Ice-Free Arctic, J. Climate, 29, 6993 – 7008, https://doi.org/10.1175/JCLI-D-15-0733.1, 2016.

Petty, A. A., Holland, M. M., Bailey, D. A., and Kurtz, N. T.: Warm Arctic, Increased Winter Sea Ice Growth?, Geophysical Research Letters, 45, 12 922–12 930, https://doi.org/10.1029/2018gl079223, 2018.

Ricker, R., Kauker, F., Schweiger, A., Hendricks, S., Zhang, J. L., and Paul, S.: Evidence for an Increasing Role of Ocean Heat in Arctic Winter Sea Ice Growth, Journal of Climate, 34, 5215–5227, https://doi.org/10.1175/jcli-d-20-0848.1, 2021.

Taylor, J. R.: An introduction to error analysis: The study of uncertainties in physical measuremenets, University Science Books, 20 Edgehill Rd., Mill Valley, CA 94941, 1982.

Tschudi, M. A., Meier, W. N., and Stewart, J. S.: An enhancement to sea ice motion and age products at the National Snow and Ice Data Center (NSIDC), Cryosphere, 14, 1519–1536, https://doi.org/10.5194/tc-14-1519-2020, 2020.

---

## Referee Report (RR1)

Following the first reviews of this paper the authors have made extensive additions to the paper. The work on presenting uncertainties to the estimates are theoretically sound and the additional figures are welcome. However, I can still not suggest this paper for publication and additional major corrections are required. As this is a second review session is it up to the editor whether to pursue this publication to a third round. At the moment the results of the paper are speculative and are lacking context with other budget estimates. Here are the major points:

While uncertainties have been calculated and discussed, all reported values and line plots within the paper have no error estimates included. The calculated estimates must be included within these figures and numbers. In particular, figures 7 and 10 need error bars, and the total numbers presented in section 4 need +/- values after each one.

The total budget values, that are the main headline results of this paper need to be contrasted to existing literature. The use of SLICE to estimate total winter thermodynamic growth is a bold but useful result. The context of how these numbers fit with existing published values need to be added to allow the results presented here to be used in any future work. For example, how do the seasonal total or averaged weekly budget values given here compare to those given by Ricker et al.? As mentioned in the previous review, the values given here for dynamic changes (or all the possible residual effects) to sea ice volume are higher than given in previous studies, with values comparable to thermodynamics. Previous work typically has dynamic changes at ¼ of thermodynamics. This is seen in Ricker et al. (2021) and also within all the models shown in Keen et al. (2021). A direct comparison between these total budget values, within the context of the given uncertainty estimates is required for the community to understand the usefulness and accuracy of these new presented results. A table putting all these values together will be helpful. Do the results presented here suggest we need to rethink where sea ice grows in the arctic? Does this paper generally agree with previous work on where ice grows? At the moment this paper just adds confusion to these questions, and this leads a reader to discount the results given here. The current presentation of uncertainty compounds this, as the given maps in figure 5 suggest that the given results are very accurate, but with no presentation of context to existing estimates, the reader is likely to doubt both these budget estimates and uncertainty estimates. The very low reported estimates for thermodynamic uncertainty adds to this (figure 5 has it at near zero). To report that a new experimental data product has near zero uncertainty is highly suspicious and leads to the conclusion that both the data and uncertainty estimates of this product are unreliable. As mentioned in the previous review, a significant aspect of this study is a presentation of the usefulness and context of the SLICE data. This aspect is currently not given enough discussion, and is not mentioned at all within the abstract.

Finally the method of only considering ice of high concentration is sensible in the context of this paper and the SLICE. However, when presenting the total budget values for a season, this study needs to include an estimate of all the volume changes that are not included when ignoring low ice concentrations. While this may be a small number, the reported thermodynamic growth in certain areas (central region) is also small, and the volume change during low sea ice concentration events may be significant.

The above point with additional considerations from line 449 in the manuscript, can be added to the explanation of the residual data. The missing low concentration contribution, plus additional lateral and new or frazil ice growth (see Keen et al. for all of these), will be apparent within the residual field.

Specific comments:
Figure 2, and then throughout, why is the season 2011-2012 not included? I guess for a technical reason, but this this needs to be clearly stated in the data or methods section.

Figure 7. This figure will benefit from the total dh/dt values as well as the components. Is the 'deformation' the same as the 'residual' given elsewhere? If so then it needs relabelling. Uncertainties need to be added (whisker plots or shaded regions). The units are confusing, the y-aixs of m/week clashes with the time period of monthly data.

Figure 10, an improved caption with all lines showing which is from this papers budget calculations and which are from other data is need. Uncertainty values need plotting too.

L 449, additionally there is new and frazil ice growth terms. See Keen et al. These are considerable and comparable to ice deformation effects in some models.

Keen, A. et al. 2021. An inter-comparison of the mass budget of the Arctic sea ice in CMIP6 models. *The Cryosphere*. 15, 2 (Feb. 2021), 951–982. DOI:https://doi.org/10.5194/tc-15-951-2021.

Ricker, R. et al. 2021. Evidence for an Increasing Role of Ocean Heat in Arctic Winter Sea Ice Growth. *Journal of Climate*. 34, 13 (Jul. 2021), 5215–5227. DOI:https://doi.org/10.1175/JCLI-D-20-0848.1.

---

## Author Response (AR2)

**Response to Referee #1**

**By Anheuser, et al.**

Author responses in red.

I thank the authors for their detailed reply on my comments. Many of my questions and comments have been addressed sufficiently. I appreciate the effort of providing a more detailed uncertainty analysis.
But I still have some remaining comments, see below.

Thanks for the additional comments, they continue to help improve the paper.

First, on your reply to my question regarding ice growth in leads:

"With a 95% sea ice concentration threshold, this will likely not be a significant source of volume generation. Nonetheless, have added a description of this issue to the discussion."
There are several papers I believe that point out the role of leads for ice growth, for example, Boutin et al. (2023), where they write: "We suggest a way to estimate the contribution of leads and polynyas to ice growth in winter, and we estimate this contribution to add up to 25 %–35 % of the total ice growth in pack ice from January to March". Of course, this is a model study, and also comes with uncertainties. But at least it seems that leads in fact play an important role for winter ice growth. I think this should be discussed in the paper and will help the reader to better understand the limitations and uncertainties here.

In the Boutin et al. (2023) methodology, the contribution from leads and polynyas is quantified as the thickness growth in the pack ice that falls in the "new ice" category, defined as any ice below 18 cm. Some of that growth will indeed be included in our thermodynamic growth and likely some of that will be included in the residual, though it is unclear how much without a more detailed analysis, which is outside the scope of this study. As such, the 25%-35% number is not an accurate estimate of how much thermodynamic growth will be missed by only considering sea ice above 95% and not accounting for sea ice concentration changes. It is only the initial closing of the open water lead that is missed in our methodology. We have added a bit to the discussion of this issue:

*It should be noted that new ice formation in leads that have opened up due to divergence in the flow field will not be included in the thermodynamic effect; rather, the balance of the new ice thickness and the thickness of the remaining ice in the grid cell will be quantified as negative residual effect. This type of effect has the potential to be greater than 5%, as the leads may grow to arbitrary size and not be apparent in the passive microwave sea ice concentration product if new ice fills the lead prior to a new passive microwave brightness temperature is taken. It is this initial closing of the lead with new ice that will not be captured as thermodynamic growth.*

L238: Regarding formula 16, I am not sure if the uncertainty of the mean is reduced in any case. Why should it? You are not averaging observations of the same subject (same piece of sea ice),

but spatially (or temporally). Yet, I am not an expert in error statistics, and I acknowledge that some assumptions have to be made here.

It may not be the same object we are observing but it is the same location for which we are determining the climatology. If the climatology is a mean of all observations at that location and the observations have random and independent uncertainties, these uncertainties reduce as additional measurements are taken. In this latest revision, Eq. 16 is applied only for temporal averaging; we found that applying this equation to the regional spatial averaging in Fig. 5 led to unreasonably low uncertainties. We are not experts in this either, merely working to most accurately characterize the uncertainty.

Figure 7: I think it would improve the readability to include a legend explaining the different lines and colors. Moreover, I would add the uncertainties here, which would provide useful information on how robust those estimate are. Otherwise the uncertainty estimate are barely used in the paper, only showing the mean fields in Fig. 5.

We have updated Fig. 7 (now Fig. 5 in the revised manuscript) the plot to a bar plot in order to better compare results with existing literature and made sure the legend is clear. Along with this, we have added uncertainty bars to this plot as well as uncertainty shading to Fig. 10.

**Response to Referee #2**

**By Anheuser, et al.**

Author responses in red.

Following the first reviews of this paper the authors have made extensive additions to the paper. The work on presenting uncertainties to the estimates are theoretically sound and the additional figures are welcome. However, I can still not suggest this paper for publication and additional major corrections are required. As this is a second review session is it up to the editor whether to pursue this publication to a third round. At the moment the results of the paper are speculative and are lacking context with other budget estimates. Here are the major points:

Thanks for the additional comments, they continue to help improve the paper.

While uncertainties have been calculated and discussed, all reported values and line plots within the paper have no error estimates included. The calculated estimates must be included within these figures and numbers. In particular, figures 7 and 10 need error bars, and the total numbers presented in section 4 need +/- values after each one.

We have added error bars to Fig. 7 (now Fig. 5 in the revised manuscript), shaded error regions to Fig. 10 and uncertainties to reported values in Section 4 where appropriate.

The total budget values, that are the main headline results of this paper need to be contrasted to existing literature. The use of SLICE to estimate total winter thermodynamic growth is a bold but

useful result. The context of how these numbers fit with existing published values need to be added to allow the results presented here to be used in any future work. For example, how do the seasonal total or averaged weekly budget values given here compare to those given by Ricker et al.? As mentioned in the previous review, the values given here for dynamic changes (or all the possible residual effects) to sea ice volume are higher than given in previous studies, with values comparable to thermodynamics. Previous work typically has dynamic changes at 1⁄4 of thermodynamics. This is seen in Ricker et al. (2021) and also within all the models shown in Keen et al. (2021). A direct comparison between these total budget values, within the context of the given uncertainty estimates is required for the community to understand the usefulness and accuracy of these new presented results. A table putting all these values together will be helpful. Do the results presented here suggest we need to rethink where sea ice grows in the arctic? Does this paper generally agree with previous work on where ice grows? At the moment this paper just adds confusion to these questions, and this leads a reader to discount the results given here. The current presentation of uncertainty compounds this, as the given maps in figure 5 suggest that the given results are very accurate, but with no presentation of context to existing estimates, the reader is likely to doubt both these budget estimates and uncertainty estimates. The very low reported estimates for thermodynamic uncertainty adds to this (figure 5 has it at near zero). To report that a new experimental data product has near zero uncertainty is highly suspicious and leads to the conclusion that both the data and uncertainty estimates of this product are unreliable. As mentioned in the previous review, a significant aspect of this study is a presentation of the usefulness and context of the SLICE data. This aspect is currently not given enough discussion, and is not mentioned at all within the abstract.

Thank you for this very helpful critique which we feel has greatly improved the paper. We have changed Fig. 7 (now Fig. 5 in the revised manuscript) to be a bar plot of volume changes and changed the regions to match those of Ricker et al. (2021) to enable a more direct comparison. We have also added a plot to this figure showing effects summed across the entire Arctic for comparison to Keen et al. (2021) and added a bar for volume changes from CS2SMOS that occur in areas below the sea ice concentration threshold. Please see below:

[Figure]

*Figure 5: Mean monthly volumetric thermodynamic growth (blue), dynamic effect (green), advection effect (orange), deformation effect (pink) and ≤95\% sea ice concentration volume changes for a) the entire Arctic, b) Central Arctic, c) Beaufort Sea, d) Chukchi Sea, e) East Siberian Sea, f) Laptev Sea, g) Kara Sea, h) Barents Sea, i) East Greenland Sea, j) Baffin Bay and k) Canadian Archipelago. Overall the entire Arctic, dynamics has a negative volume effect that is -25% that of thermodynamic growth.*

What we see is that the regional and basin-wide results in fact agree well with Ricker et al. (2021) and Keen et al. (2021). The new information offered by this work does not require a rethinking of sea ice dynamics but rather adds new spatial detail showing some regions deviate significantly from these mean values.

As a part of developing this new figure, we have revisited our uncertainty methodology. To address unreasonably large uncertainties in the dynamic components of this plot, we have added covariances to the time and spatial derivatives of CS2SMOS. We have added the following to Section 3.1 in addition to updated the uncertainty equation and Fig. 4:

*The uncertainty in the space and time derivatives of CS2SMOS contain covariance terms. CS2SMOS uncertainty is a significant source of uncertainty within the uncertainty framework above but some portion of this uncertainty would cancel when a difference between time steps or neighboring grid points is taken. Though these covariances have not been explored in literature in relation to CS2SMOS, we look to Fig. 7 within Ricker et al. (2017b) for guidance on correlation between grid cells within a single CS2SMOS field. For the example region depicted in this figure, correlation between thickness observations at grid points located less than 100 km apart are nearly always greater than 0.6. This 100 km radius includes neighboring grid points which are separated by 25 km and displacement during the two weeks between time steps in Eq. 6, which typically won't exceed 100 km. Based on this figure, we assume a correlation between time steps or neighboring grid points of 0.6 as a conservative estimate. Other than in this instance, our uncertainty formulas do not account for covariances between the input terms. Though covariances may be present across the input data, inclusion of their effects on uncertainty is outside the scope of this work.*

Finally the method of only considering ice of high concentration is sensible in the context of this paper and the SLICE. However, when presenting the total budget values for a season, this study needs to include an estimate of all the volume changes that are not included when ignoring low ice concentrations. While this may be a small number, the reported thermodynamic growth in certain areas (central region) is also small, and the volume change during low sea ice concentration events may be significant.

As mentioned above, we have added a bar to Fig. 7 (now Fig. 5 in the revised manuscript) showing, for each region and the entire Arctic, how much volume growth occurs below the sea ice concentration threshold per CS2SMOS. Across the entire Arctic, this volume growth is 13% that of thermodynamic growth.

The above point with additional considerations from line 449 in the manuscript, can be added to the explanation of the residual data. The missing low concentration contribution, plus additional lateral and new or frazil ice growth (see Keen et al. for all of these), will be apparent within the residual field.

We have added the following to the methodology:

*The residual difference of the overall dynamic effect and this advection effect includes the effects of ice deformation and any other effects that are not accounted for in SLICE or the calculation of advection. These additional effects include lateral growth, snow ice formation and any frazil or new ice growth that occurs above 95% sea ice concentration and is not captured within SLICE.*

Specific comments:
Figure 2, and then throughout, why is the season 2011-2012 not included? I guess for a technical reason, but this this needs to be clearly stated in the data or methods section.

This was mentioned in the methodology Section is found at line 214.

*The 2010 data begins on 15 November rather than 1 November along with the availability of CryoSat-2 data and the winter beginning in 2011 is not included due to a gap between availability of passive microwave data from the earlier AMSR-E and latter AMSR2.*

Figure 7. This figure will benefit from the total dh/dt values as well as the components. Is the 'deformation' the same as the 'residual' given elsewhere? If so then it needs relabelling. Uncertainties need to be added (whisker plots or shaded regions). The units are confusing, the y-aixs of m/week clashes with the time period of monthly data.

Please see above explanation of changes to Fig. 7 (now Fig. 5 in the revised manuscript). Additionally, we have updated all figures to be in a m/month unit. We agree this makes the results more streamlined and intuitive.

Figure 10, an improved caption with all lines showing which is from this papers budget calculations and which are from other data is need. Uncertainty values need plotting too.

We have added "from our results" where appropriate in this caption and added uncertainty shading.

L 449, additionally there is new and frazil ice growth terms. See Keen et al. These are considerable and comparable to ice deformation effects in some models.

We have updated this sentence to:

*The SLICE thermodynamic growth retrieval also does not account for lateral melt and freeze processes, or any new or frazil ice growth that occurs above 95% sea ice concentration.*

Keen, A. et al. 2021. An inter-comparison of the mass budget of the Arctic sea ice in CMIP6 models. *The Cryosphere*. 15, 2 (Feb. 2021), 951–982. DOI:https://doi.org/10.5194/tc-15-951-2021.

Ricker, R. et al. 2021. Evidence for an Increasing Role of Ocean Heat in Arctic Winter Sea Ice Growth. *Journal of Climate*. 34, 13 (Jul. 2021), 5215–5227. DOI:https://doi.org/10.1175/JCLI-D-20-0848.1.